# TOWARDS IDENTIFIABLE LATENT ADDITIVE NOISE MODELS

## ABSTRACT

Causal representation learning (CRL) offers the promise of uncovering the underlying causal model by which observed data was generated, but the practical applicability of existing methods remains limited by the strong assumptions required for identifiability and by challenges in applying them to real-world settings. Most current approaches are applicable only to relatively restrictive model classes, such as linear or polynomial models, which limits their flexibility and robustness in practice. One promising approach to this problem seeks to address these issues by leveraging changes in causal influences among latent variables. In this vein we propose a more general and relaxed framework than typically applied, formulated by imposing constraints on the function classes applied. Within this framework, we establish partial identifiability results under weaker conditions, including scenarios where only a subset of causal influences change. We then extend our analysis to a broader class of latent post-nonlinear models. Building on these theoretical insights, we develop a flexible method for learning latent causal representations. We demonstrate the effectiveness of our approach on synthetic and semi-synthetic datasets, and further showcase its applicability in a case study on human motion analysis, a complex real-world domain that also highlights the potential to broaden the practical reach of identifiable CRL models.

## 1 INTRODUCTION

Causal representation learning (CRL) aims to recover the latent variables and causal structures that give rise to high-dimensional observations, offering a principled perspective on modeling complex systems (Schölkopf et al., 2021; Ahuja et al., 2023). By explicitly capturing underlying generative mechanisms, CRL enhances interpretability and supports robust generalization across environments, particularly under distribution shifts induced by interventions (Peters et al., 2017; Pearl, 2000). Such capabilities make CRL particularly valuable in domains such as reinforcement learning and self-supervised learning, where uncovering latent causal factors can facilitate more general yet effective representations and enable more effective planning (Mitrovic et al., 2021; Zeng et al., 2024). While CRL holds clear advantages over correlation-based methods, yielding representations that are more robust and transferable, it remains difficult to realize these benefits in practice. This is largely due to the strong assumptions required for identifiability and the challenges associated with deploying existing models in realistic environments (Bing et al., 2024; Yao et al., 2025).

Leveraging changes in causal influences among latent variables has emerged as a promising strategy to enhance identifiability and improve estimation quality. Recent work in this direction has focused on developing theoretically grounded frameworks for identifiability, alongside practical methods tailored for real-world applications (Squires et al., 2023; Liu et al., 2022; Buchholz et al., 2023; Liu et al., 2024; Von Kügelgen et al., 2021; Brehmer et al., 2022; Ahuja et al., 2023; Varici et al., 2023; von Kügelgen et al., 2023). Underlying these methods is the core intuition that changes in causal influences introduce asymmetries, e.g., pre- and post-change behaviors, into the system, which provides valuable signals that help achieve identifiability. Based on this idea, prior works have established various identifiability results for restricted function classes over latent variable models, including, but not limited to linear Gaussian models (Liu et al., 2022; Buchholz et al., 2023), linear additive noise models (Squires et al., 2023; Chen et al., 2024; Jin & Syrgkanis, 2024), and polynomial models (Liu et al., 2024). More related work can be found in Sec. A. A deeper comparison with polynomial models in Liu et al. (2024) is provided in Sec. P.

**Remaining Challenges.** Despite this progress, several limitations pose difficulties for broader applicability. Theoretically, many existing identifiability results rely on strong assumptions, such as specific functional forms or distributional constraints, which may not hold in complex or poorly understood real-world systems. Empirically, due to the relatively strong assumption required for changes in causal influences in latent space, real-world applications that conform to these assumptions remain limited. As a result, many identifiability results are primarily evaluated on synthetic datasets, typically Causal3DIdent (Von Kügelgen et al., 2021). Although there have been promising attempts to adapt these identifiability results into effective methods for real-world data, particularly in biological data (Squires et al., 2023; Zhang et al., 2023) and climate data (Yao et al., 2024), further efforts are needed to extend CRL to a wider range of real-world scenarios.

**Contributions.** In this work, we aim to advance the study of leveraging changes in causal influences by contributing to both theory and practical applications. *On the theoretical side*, we introduce a nonparametric condition that characterizes changes in causal influences between latent causal variables. Under this condition and standard assumptions from nonlinear ICA (Hyvarinen & Morioka, 2016; Hyvarinen et al., 2019; Khemakhem et al., 2020), we show that general latent additive noise models can be identified up to permutation and scaling. We further extend this result to a more realistic setting where only a subset of the latent causal variables undergoes changes, resulting in partial identifiability. ~~Notably, our analysis shows that the proposed nonparametric condition is both necessary and sufficient for identifiability under the nonlinear ICA framework, without additional constraints~~. We also generalize these results to latent post-nonlinear models, which include additive noise models as a special case. *On the practical side*, we explore a novel real-world application: learning causal representations from human motion data. In this setting, the underlying latent causal system can be interpreted as dynamic motor control modules that govern human motion across different tasks (Gallego et al., 2017; Doyon & Benali, 2005; Taylor et al., 2006). This opens a new potential application for applying causal representation learning to complex human-centered data.

## 2 LATENT ADDITIVE NOISE MODELS WITH CHANGE IN CAUSAL INFLUENCES

We consider a general class of latent additive noise models, where the observed data $\mathbf{x}$ is generated from a set of latent causal variables $\mathbf{z} \in \mathbb{R}^{\ell}$. These latent causal variables are causally influenced by latent noise terms $\mathbf{n} \in \mathbb{R}^{\ell}$, and their causal relationships are represented by a directed acyclic graph (DAG). Importantly, we do not assume a fixed graph structure over the latent causal variables, allowing for flexible modeling of their dependencies and applicability across different settings. To account for the change of causal influences between latent causal variables $\mathbf{z}$, which may arise from environmental or contextual factors, we introduce a surrogate variable $\mathbf{u}$. This variable plays a central role in capturing how changes in external conditions are reflected in the observed data $\mathbf{x} \in \mathbb{R}^{\ell}$. The interpretation of $\mathbf{u}$ is application-dependent. In domain adaptation or generalization tasks, it may represent environmental factors that vary across domains. In time series forecasting (Mudelsee, 2019), $\mathbf{u}$ can capture temporal indices reflecting evolving trends. In remote sensing (Rußwurm et al., 2020), it may encode geographic attributes such as longitude and latitude that influence observations.

**Data Generation Process.** More specifically, we parameterize the latent causal generative models by assuming $\mathbf{n}$ follows an exponential family distribution given $\mathbf{u}$, and $\mathbf{z}$ and $\mathbf{x}$ are generated as follows:

$$p_{(\mathbf{T}, \boldsymbol{\eta})}(\mathbf{n} \mid \mathbf{u}) := \prod_i \frac{\exp\left(\sum_j T_{i,j}(n_i)\, \eta_{i,j}(\mathbf{u})\right)}{Z_i(\mathbf{u})}, \quad (1) \qquad z_i := \mathrm{g}_i^{\mathbf{u}}(\mathrm{pa}_i) + n_i, \quad (2) \qquad \mathbf{x} := \mathbf{f}(\mathbf{z}) + \boldsymbol{\varepsilon}. \quad (3)$$

In Eq. (1): $Z_i(\mathbf{u})$ is the normalizing constant, and $T_{i,j}(n_i)$ is the sufficient statistic for $n_i$, with its natural parameter $\eta_{i,j}(\mathbf{u})$ dependent on $\mathbf{u}$. We assume a two-parameter exponential family, following the formulation in Sorrenson et al. (2020). In Eq. (2): The term $\mathrm{g}_i^{\mathbf{u}}(\mathrm{pa}_i)$ shows how $\mathbf{u}$ influences the mapping of parents $\mathrm{pa}_i$ to $z_i$. Specifically, $\mathbf{u}$ modulates the function $\mathrm{g}_i$, e.g., if $\mathrm{g}_i$ is modeled by a multilayer perceptron (MLP), $\mathbf{u}$ adjusts the weights of the MLP. In Eq. (3): $\mathbf{f}$ represents a nonlinear mapping from $\mathbf{z}$ to $\mathbf{x}$, where $\boldsymbol{\varepsilon}$ is independent noise with density function $p_{\boldsymbol{\varepsilon}}(\boldsymbol{\varepsilon})$, with $\boldsymbol{\varepsilon} \in \mathbb{R}^{\ell}$.

The surrogate variable $\mathbf{u}$ captures distributional shifts in the latent noise variables, as reflected by changes in the natural parameters across $\mathbf{u}$ in Eq.(1). This enables the adaptation of identifiability results from nonlinear ICA. More importantly, $\mathbf{u}$ also models changes in the underlying causal influences from parent variables to each latent causal variable. In particular, as shown in Eq.(2), $\mathbf{u}$ modulates the functional form $\mathrm{g}_i^{\mathbf{u}}$, effectively characterizing how the parents $\mathrm{pa}_i$ influence $z_i$. By

imposing appropriate constraints on $g_i^{\mathbf{u}}$, we can identify a sufficient ~~and necessary~~ condition for changes in the causal influences among latent variables (see assumption (iv)), under nonlinear ICA.

## 3 IDENTIFIABILITY RESULTS OF LATENT ADDITIVE NOISE MODELS

In this section, we analyze identifiability in latent additive noise models by leveraging changes in causal influences across environments. We first present the complete identifiability result inSection 3.1, then extend to partial identifiability result in Section 3.2, addressing more general and realistic scenarios in which only a subset of causal influences among latent variables undergo change. Finally, we generalize both complete and partial results to latent post-nonlinear models in Section 3.3.

### 3.1 COMPLETE IDENTIFIABILITY RESULT OF LATENT ADDITIVE NOISE MODELS

We explicitly introduce the surrogate variable $\mathbf{u}$ as described in the data generation process defined by Eqs. (1)–(3). This mechanism allows us to formulate key identifiability conditions in terms of $\mathbf{u}$. We now present the main identifiability result as follows:

**Theorem 3.1.** *Suppose latent causal variables $\mathbf{z}$ and the observed variable $\mathbf{x}$ follow the causal data generative models defined in Eqs. (1) - (3). Assume the following holds:*

*(i)* ~~*The noise probability density function $p_{\boldsymbol{\varepsilon}}(\boldsymbol{\varepsilon})$ does not depend on $\mathbf{u}$ and is always finite,*~~ *The set $\{\mathbf{x} \in \mathcal{X} | \varphi_{\varepsilon}(\mathbf{x}) = 0\}$ has measure zero (i.e., has at most countable number of elements), where $\varphi_{\boldsymbol{\varepsilon}}$ is the characteristic function of the density $p_{\boldsymbol{\varepsilon}}$,*

*(ii)* *The function $\mathbf{f}$ in Eq. (3) is smooth and invertible,*

*(iii)* *There exist $2\ell + 1$ values of $\mathbf{u}$, i.e., $\mathbf{u}_0, \mathbf{u}_1, ..., \mathbf{u}_{2\ell}$, such that the matrix*

$$\mathbf{L} = (\boldsymbol{\eta}(\mathbf{u} = \mathbf{u}_1) - \boldsymbol{\eta}(\mathbf{u} = \mathbf{u}_0), ..., \boldsymbol{\eta}(\mathbf{u} = \mathbf{u}_{2\ell}) - \boldsymbol{\eta}(\mathbf{u} = \mathbf{u}_0)) \tag{4}$$

*of size $2\ell \times 2\ell$ is invertible, where $\boldsymbol{\eta}(\mathbf{u}) = [\eta_{i,j}(\mathbf{u})]_{i,j}$,*

*(iv)* *The function class of $g_i^{\mathbf{u}}$ satisfies the following condition: there exists $\mathbf{u}_i$, such that, for all parent nodes $z_j \in \mathrm{pa}_i$ of $z_i$, $\frac{\partial g_i^{\mathbf{u}=\mathbf{u}_i}(\mathrm{pa}_i)}{\partial z_j} = 0$.*

*Then each true latent variable $z_i$ is linearly related to exactly one estimated latent variable $\hat{z}_j$, as $z_i = s_j \hat{z}_j + c_i$, for some constants $s_j$ and $c_i$, where all $\hat{z}_j$ are learned by matching the true data distribution $p(\mathbf{x} \mid \mathbf{u})$.*

**Proof sketch** First, we show that the latent noise variables are identifiable up to scaling and permutation, leveraging recent progress in nonlinear ICA (supported by assumptions (i)–(iii)) and the structure of additive noise models (Eq. 2, Lemma B.2). Second, building on this, the true latent causal variables $\mathbf{z}$ are related to the estimated ones through an invertible map that is independent of the auxiliary variable $\mathbf{u}$, by the additive noise model structure (Lemma B.1). Finally, using the changes in causal influences among $\mathbf{z}$ (assumption (iv)), this invertible map reduces to permutation and scaling. Full details are provided in Appendix C.

Assumptions (i)-(iii) are originally developed by nonlinear ICA (Hyvarinen & Morioka, 2016; Hyvarinen et al., 2019; Khemakhem et al., 2020; Sorrenson et al., 2020), and have also been adopted in several recent works on CRL (may with different forms) Liu et al. (2022; 2024); Zhang et al. (2024); Ng et al. (2025). It is worth noting that the polynomial setting in Liu et al. (2024) can be seen as a special case of our *additive noise models*, with assumption (iv) providing the key identifiability condition (See Appendix P for a detailed comparison).. We here unitize these assumptions considering the following two main reasons. 1) These assumptions have been verified to be practicable in diverse real-world application scenarios (Kong et al., 2022; Xie et al., 2022b; Wang et al., 2022). 2) By extending the results of Sorrenson et al. (2020) to our setting[1], we can identify the latent noise variables up to permutation and scaling, which in turn facilitates the identifiability of the latent causal

---

[1]This extension requires addressing a technical gap introduced by Eq. (2), specifically ensuring that (i) the mapping from $\mathbf{n}$ to $\mathbf{x}$ remains invertible, and (ii) $\mathbf{u}$ does not compromise the identifiability of $\mathbf{n}$.

variables. ~~it suffices to know that the number of environments exceeds $2\ell + 1$, which is somewhat more lenient than requiring the exact number, as is the case in some prior works.~~

Assumption (iv), originally introduced by this work, provides a condition that characterizes the types of change in causal influences contributing to identifiability. Loosely speaking, this assumption ensures that the causal influence from parent nodes does not include components that remain invariant across $\mathbf{u}$, as such invariance would lead to unidentifiability (See Remark 3.2 for more details). This is achieved by constraining the gradient of $g_i^{\mathbf{u}}$ with respect to $z_j$ vanished at the point $\mathbf{u}_i$, thereby preventing the invariance. From a high-level perspective, this closely aligns with the notion of perfect interventions discussed in prior works (von Kügelgen et al., 2023; Buchholz et al., 2023; Wendong et al., 2023), thereby ensuring no terms that link $z_i$ and its parent node remain unchanged.

Assumption (iv), for instance, could arise in the analysis of cell imaging data (i.e., $\mathbf{x}$), where various batches of cells are exposed to different small-molecule compounds (i.e., $\mathbf{u}$). each latent variable (i.e., $z_i$) represents the concentration level of a distinct group of proteins, with protein-protein interactions (e.g., causal influences among $z_i$) playing a significant role (Chandrasekaran et al., 2021). Research has revealed that the mechanisms of action of small molecules exhibit variations in selectivity (Scott et al., 2016), which can profoundly affect protein-protein interactions (i.e., $g_i^{\mathbf{u}}$). The assumption (iv) requires the existence of a specific $\mathbf{u} = \mathbf{u}_i$, such that the original causal influences can be disconnected. This parallels cases where small molecule compounds disrupt or inhibit protein-protein interactions (PPIs), effectively causing these interactions to cease (Arkin & Wells, 2004). Such molecules are commonly referred to as inhibitors of PPIs. Developing small molecule inhibitors for PPIs is a key focus in drug discovery (Lu et al., 2020; Bojadzic et al., 2021). Additionally, gene editing technologies like CRISPR/Cas9 can effectively 'knock out' a protein or gene, leading to complete inhibition. Similarly, receptor antagonists can achieve full inhibition by completely blocking the activity of a receptor.

We emphasize that assumption (iv) is our key contribution, formulating changes in causal influence as constraints on the function class and thus distinguishing our work from previous studies. Specifically,

**Remark 3.2** (Types of Changes in Causal Influences That Facilitate Identifiability). *Not all changes in causal influences lead to identifiability. Assumption (iv) specifies the types of changes in causal influences among latent causal variables contributing to identifiability.*

To clarify this point, consider the following example.

**Example 3.3.** Let $z_2 := \mathrm{MLP}^{\mathbf{u}}(z_1) + n_2$, where $\mathrm{MLP}^{\mathbf{u}}(z_1)$ can be decomposed as $\mathrm{MLP}^{\mathbf{u}}(z_1) = \mathrm{MLP}_1^{\mathbf{u}}(z_1) + \mathrm{MLP}_2(z_1)$, with $\mathrm{MLP}_1^{\mathbf{u}}(z_1)$ being the $\mathbf{u}$-dependent component and $\mathrm{MLP}_2(z_1)$ being a $z_1$-dependent term invariant across $\mathbf{u}$. Both $\mathrm{MLP}^{\mathbf{u}}$ and $\mathrm{MLP}_1^{\mathbf{u}}$ belong to the same function class.

In this example, if assumption (iv) is violated, $z_2$ becomes unidentifiable. While the causal influence from $z_1$ to $z_2$ changes across $\mathbf{u}$ due to the $\mathbf{u}$-dependent term $\mathrm{MLP}_1^{\mathbf{u}}(z_1)$, the invariant term $\mathrm{MLP}_2(z_1)$ induces a invariant causal link between $z_1$ and $z_2$ across $\mathbf{u}$, which leads to unidentifiable result. Specifically, the invariant $\mathrm{MLP}_2(z_1)$ can be absorbed into the generative mapping $\mathbf{f}$, resulting in an alternative representation $z_2' := \mathrm{MLP}_1^{\mathbf{u}}(z_1) + n_2$, which would generate the same observational data. That is, the original data generation process can be equivalently written as $\mathbf{x} = \mathbf{f}(z_1, z_2) = \mathbf{f} \circ \mathbf{f}_1(z_1, z_2')^2$, where $\mathbf{f}_1(z_1, z_2') = [z_1, z_2' + \mathrm{MLP}_2(z_1)]$. Consequently, the model remains unidentifiable. A formal statement of this result is given in Theorem 3.4(b). The reason of this unidentifiability is the presence of the invariant $\mathrm{MLP}_2(z_1)$, which maintains a constant causal influence of $z_1$ on $z_2$ across $\mathbf{u}$. Assumption (iv) mitigates this issue by eliminating such invariant component. It does so by constraining the function class satisfies: $\partial \mathrm{MLP}^{\mathbf{u}=\mathbf{u}_2}(z_1)/\partial z_1 = 0$ and $\partial \mathrm{MLP}_1^{\mathbf{u}=\mathbf{u}_2}(z_1)/\partial z_1 = 0$. As a result, $\mathrm{MLP}_2(z_1)/\partial z_1 = 0$, which implies that $\mathrm{MLP}_2(z_1)$ must be a constant, removing $z_1$-dependent term and thus ensuring identifiability.

## 3.2 PARTIAL IDENTIFIABILITY RESULT OF LATENT ADDITIVE NOISE MODELS

In practice, satisfying assumption (iv) for every causal influence from parent nodes to each child node can be challenging. When this assumption is violated for some nodes, full identifiability may not be achievable. Nevertheless, we can still derive partial identifiability results, as detailed below:

---

[2] For simplicity, we here omit the noise term $\varepsilon$. This omission does not affect the following analysis.

**Theorem 3.4.** *Suppose latent causal variables* $\mathbf{z}$ *and the observed variable* $\mathbf{x}$ *follow the causal generative models defined in Eqs.* (1) *-* (3)*, and the assumptions (i)-(iii) are satisfied, for each* $z_i$,

    *(a) if condition (iv) is satisfied, then the true* $z_i$ *is related to the recovered one* $\hat{z}_j$*, obtained by matching the true marginal data distribution* $p(\mathbf{x}|\mathbf{u})$*, by the following relationship:* $z_i = s_j \hat{z}_j + c_j$*, where* $s_j$ *denotes scaling,* $c_j$ *denotes a constant,*

    *(b) if condition (iv) is not satisfied, then* $z_i$ *is unidentifiable.*

**Proof sketch** As outlined in the proof sketch for Theorem 3.1, in the second step, without invoking assumption (iv), we can establish an invertible mapping between the true latent causal variables $\mathbf{z}$ and the estimated ones. Building on this mapping, when the condition in (a) is satisfied, we can directly prove (a). Conversely, for (b), we can establish the proof of (b), by removing the terms corresponding to the unchanged causal influences, as illustrated in Example 3.3. Full details are provided in Appendix D.

**Remark 3.5.** ~~[Sufficiency and Necessity of condition (iv)] The contrapositive of Theorem 3.4 (b), which asserts that if $z_i$ is identifiable, then condition (iv) is satisfied, serves to establish the necessity of condition (iv) for achieving complete identifiability. This insight, coupled with Theorem 3.1, underscores that condition (iv) is not only sufficient but also necessary for the identifiability result, under assumptions (i)-(iii), without additional assumptions.~~

**Remark 3.6** (Parent nodes do not impact children)**.** The implications of Theorem 3.4 ((a) and (b)) suggest that $z_i$ remains identifiable, even when its parent nodes are unidentifiable. This is primarily because regardless of whether assumption (iv) is met, assumptions (i)-(iii) ensure that latent noise variables $\mathbf{n}$ can be identified. In the context of additive noise models (or post-nonlinear models discussed in the next section), the mapping from $\mathbf{n}$ to $\mathbf{z}$ is invertible. Therefore, with identifiable noise variables, all necessary information for recovering $\mathbf{z}$ is contained within $\mathbf{n}$. Furthermore, assumption (iv) is actually transformed into relations between each node and the noise of its parent node, as stated in Lemma B.3. As a result, $z_i$ could be identifiable, even when its parent nodes are unidentifiable.

**Remark 3.7** (Subspace identifiability)**.** The implications of Theorem 3.4 suggest the theoretical possibility of partitioning the entire latent space into two distinct subspaces: latent invariant space containing *invariant* latent causal variables and latent *variant* space comprising variant latent causal variables. This insight could be particularly valuable for applications that prioritize learning invariant latent variables to adapt to changing environments, such as domain adaptation or generalization (Kong et al., 2022). While similar findings have been explored in latent polynomial models in (Liu et al., 2024), this work demonstrates that such results also apply to more flexible additive noise models.

**Summary** This work decomposes causal mechanisms in latent space into two components: one associated with latent noise variables and the other capturing causal influences from parent nodes. By analyzing the changes of the distributions of latent noise variables, formalized by assumption (iii) in Theorem 3.1, we show that the latent noise variables $\mathbf{n}$ an be identified. However, identifying $\mathbf{n}$ alone does not ensure component-wise identifiability of the latent causal variables $\mathbf{z}$, as demonstrated by Theorem 3.4 (b). To address this, we further examine changes in the causal influences. Specifically, $\mathbf{g}_i^{\mathbf{u}}$ in Eq. (2), assumption (iv) has been proven to be a sufficient ~~and necessary~~ condition for component-wise identifiability of $\mathbf{z}$, supported by Theorem 3.1 and Theorem (a), under assumptions (i)-(iii). Finally, we extend our theory to a more practical setting where only a subset of the latent variables satisfies assumption (iv). In this case, we achieve partial identifiability, as shown in Theorem 3.4 (a).

## 3.3 EXTENSION TO LATENT POST-NONLINEAR MODELS

While latent additive noise models, as defined in Eq. (2), are general, their capacities are still limited, e.g., requiring additive noise. In this section, we generalize latent additive noise models to latent post-nonlinear models (Zhang & Hyvärinen, 2009; Uemura et al., 2022; Keropyan et al., 2023), which generally offer more powerful expressive capabilities than latent additive noise models. To this end, we replace Eq. (2) by the following:

$$\bar{z}_i := \bar{g}_i(z_i) = \bar{g}_i(g_i^{\mathbf{u}}(\mathrm{pa}_i) + n_i), \tag{5}$$

where $\bar{g}_i$ denotes a invertible post-nonlinear mapping. It includes the latent additive noise models Eq. (2) as a special case in which the nonlinear distortion $\bar{g}_i$ does not exist. Based on this, we can identify $\bar{\mathbf{z}}$ up to component-wise invertible nonlinear transformation as follows:

**Corollary 3.8.** *Suppose latent causal variables* $\mathbf{z}$ *and the observed variable* $\mathbf{x}$ *follow the causal generative models defined in Eqs.* (1), (5) *and* (3). *Assume that conditions (i) - (iv) in Theorem 3.1 hold, then each true latent variable* $\bar{z}_i$ *is related to exactly one estimated latent variable* $\hat{\bar{z}}_j$, *which is learned by matching the true marginal data distribution* $p(\mathbf{x}|\mathbf{u})$, *by the following relationship:* $\bar{z}_i = M_j(\hat{\bar{z}}_j) + c_j$, *where* $M_j$ *and* $c_j$ *denote a invertible nonlinear mapping and a constant, respectively.*

**Proof sketch** The proof proceeds intuitively as follows. Since each function $\bar{g}_i$ in Eq. (5) is invertible, we can define a new invertible function by composing $\mathbf{f}$ with $\bar{\mathbf{g}}$, component-wise via $\bar{g}_i$. This composition preserves invertibility and allows us to directly apply Theorem 3.1, yielding the stated identifiability result. Full details are provided in Appendix E..

Similar to Theorem 3.4, we have partial identifiability result:

**Corollary 3.9.** *Suppose latent causal variables* $\mathbf{z}$ *and the observed variable* $\mathbf{x}$ *follow the causal generative models defined in Eqs.* (1), (5) *and* (3). *Under the condition that the assumptions (i)-(iii) are satisfied, for each* $\bar{z}_i$, *(a) if it is a root node or condition (iv) is satisfied, then the true* $\bar{z}_i$ *is related to the recovered one* $\hat{\bar{z}}_j$, *obtained by matching the true marginal data distribution* $p(\mathbf{x}|\mathbf{u})$, *by the following relationship:* $\bar{z}_i = M_j(\hat{\bar{z}}_j) + c_j$, *where* $M_j$ *denotes a invertible mapping,* $c_j$ *denotes a constant, (b) if condition (iv) is not satisfied, then* $\bar{z}_i$ *is unidentifiable.*

**Proof sketch** Again, since the function $\bar{g}_i$ is invertible defined in Eq. (5) and $\mathbf{f}$ is invertible in theorem 3.4, we can use the result of theorem 3.4 (b) to conclude the proof. Refer to Appendix F.

**Remark 3.10** (Sharing Properties)**.** Corollary 3.9 establishes that the properties outlined in Theorem 3.4, including remark 3.5 to 3.7, remain applicable in latent post-nonlinear causal models.

## 4 LEARNING LATENT ADDITIVE NOISE MODELS

In this section, we translate our theoretical findings into a novel method for learning latent causal models. Our primary focus is on learning additive noise models, as extending the method to latent post-nonlinear models is straightforward, simply involving the utilization of invertible nonlinear mappings. Following previous work in (Liu et al., 2022), due to permutation indeterminacy in latent space, we can naturally enforce a causal order $z_1 > z_2 > ..., > z_\ell$ without specific semantic information. This does not imply that we require knowledge of the true causal order, refer to Appendix G for more details. With guarantee from Theorem 3.1, each variable $z_i$ can be imposed to learn the corresponding latent variables in the correct causal order. As a result, we formulate a prior model as follows:

$$p(\mathbf{z}|\mathbf{u}) = \prod_{i=1}^{\ell} p(z_i|\mathbf{z}_{<i} \odot \mathbf{m}_i(\mathbf{u}), \mathbf{u}), = \prod_{i=1}^{\ell} \mathcal{N}(\mu_{z_i}(\mathbf{z}_{<i} \odot \mathbf{m}_i(\mathbf{u}), \mathbf{u}), \delta_{z_i}^2(\mathbf{z}_{<i} \odot \mathbf{m}_i(\mathbf{u}), \mathbf{u})), \quad (6)$$

where we focus on latent Gaussian noise variables, to satisfy the exponential family assumption in Eq. (1) and naturally allow the implementation of the reparameterization trick. Moreover, we introduce additional vectors $\mathbf{m}_i(\mathbf{u})$, by enforcing sparsity on $\mathbf{m}_i(\mathbf{u})$ and the component-wise product $\odot$, which aligns with assumption (iv) and facilitates learning the latent causal graph structure. In our implementation, we impose the L1 norm, though other methods may also be flexible, e.g., sparsity priors (Carvalho et al., 2009; Liu et al., 2019). We employ a variational posterior to approximate the true posterior $p(\mathbf{z}|\mathbf{x}, \mathbf{u})$:

$$q(\mathbf{z}|\mathbf{u},\mathbf{x}) = \prod_{i=1}^{\ell} q(z_i|\mathbf{z}_{<i} \odot \mathbf{m}_i, \mathbf{u}, \mathbf{x}), = \prod_{i=1}^{\ell} \mathcal{N}(\mu_{z_i}(\mathbf{z}_{<i} \odot \mathbf{m}_i(\mathbf{u}), \mathbf{u}, \mathbf{x}), \delta_{z_i}^2(\mathbf{z}_{<i} \odot \mathbf{m}_i(\mathbf{u}), \mathbf{u}, \mathbf{x})), \quad (7)$$

where the variational posterior shares the same parameter $\mathbf{m}_i$ to limit both the prior and the variational posterior, maintaining the same latent causal graph structure. In addition, we enforce a Gaussian posterior conditioned on the parent nodes, to align with the prior model and model assumption in Eqs. (1) and (2). Finally, we arrive at the objective:

$$\max \mathbb{E}_{q(\mathbf{z}|\mathbf{x},\mathbf{u})}(\log p(\mathbf{x}|\mathbf{z}, \mathbf{u})) - D_{KL}(q(\mathbf{z}|\mathbf{x},\mathbf{u})\|p(\mathbf{z}|\mathbf{u})) - \gamma \sum_i \|\mathbf{m}_i(\mathbf{u})\|_1^1, \quad (8)$$

where $D_{KL}$ denotes the KL divergence, $\gamma$ denotes a hyperparameters to control the sparsity of latent causal structure. The objective is known as the evidence lower bound (ELBO), which serves as a

lower bound of the log-likelihood. Under certain conditions, it can match the true data distribution, which is one of the requirements for our identifiability guarantee in Theorem 3.1. Moreover, the variational estimator is consistent in the sense that, as the number of data samples grows, the learned variational posterior converges to the true posterior [3], ensuring reliable recovery of the underlying latent variables. Implementation details can be found in Appendix J.

## 5 EXPERIMENTS

**Synthetic Data** We first conduct experiments on synthetic data, generated by the following process: we divide latent noise variables into $M$ segments, where each segment corresponds to one value of $\mathbf{u}$ as the segment label. Within each segment, the location and scale parameters are respectively sampled from uniform priors. After generating latent noise variables, we generate latent causal variables, and finally obtain the observed data samples by an invertible nonlinear mapping on the causal variables. Details can be found in H.

We evaluate our proposed method, implemented with multilayer percep-trons (MLPs) and hence referred to as MLPs, to model the causal rela-tions among latent causal variables, against established models: vanilla VAE (Kingma & Welling, 2013), $\beta$-VAE (Higgins et al., 2017), identifi-able VAE (iVAE) (Khemakhem et al., 2020), and latent polynomial models (Polynomials) (Liu et al., 2024). No-tably, the iVAE demonstrates the ca-pability to identify true independent noise variables, subject to certain con-ditions, with permutation and scaling.

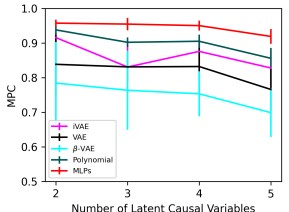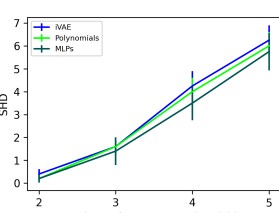

Figure 1: Performance comparison under latent additive Gaus-sian noise. Left: MPC scores for different methods, where the proposed MLPs method achieves the best performance, supporting our theoretical results. Right: SHD scores of the proposed method, Polynimals (Liu et al., 2024), and iVAE combined with the method from Huang et al. (2020).

Polynomials, while sharing similar assumptions with our proposed method, are prone to certain limitations. Specifically, they may suffer from numerical instability and face challenges due to the exponential growth in the number of terms. While the $\beta$-VAE is popular in disentanglement tasks due to its emphasis on independence among recovered variables, it lacks robust theoretical backing. Our evaluation focuses on two metrics: the Mean of the Pearson Correlation Coefficient (MPC) to assess performance, and the Structural Hamming Distance (SHD) to gauge the accuracy of the latent causal graphs. The result for iVAE is obtained by applying the method from (Huang et al., 2020) to the latent variables estimated by iVAE. Figure 1 illustrates the comparative performances of various methods, e.g., VAE and iVAE, across different models, e.g., models with different dimensions of latent variables. Based on MPC, the proposed method demonstrates satisfactory results, thereby supporting our identifiability claims. Additionally, Figure 2 presents how the proposed method performs when condition (iv) is not met. It is evident that condition (iv) is a sufficient ~~and necessary~~ condition characterizing the types of distribution shifts for identifiability in the context of latent additive noise models. These empirical findings align with our partial identifiability results. More results on high-dimensional synthetic image data can be found in Appendix L.

**Post-Nonlinear Models** In the above experiments, we obtain the observed data samples as derived from a random invertible nonlinear mapping applied to the latent causal variables. The nonlinear mapping can be conceptualized as a combination of an invertible transformation and the specific invertible mapping, $\bar{g}_i$. From this perspective, the results depicted in Figures 1 and 2 also demonstrate the effectiveness of the proposed method in recovering the variables $z_i$ in latent post-nonlinear models Eq. (5), as well as the associated latent causal structures. Consequently, these results also serve to corroborate the assertions in Corollary 3.8 and 3.9, particularly given that $\bar{g}_i$ are invertible.

---

[3]More strictly, this holds under the assumption that the variational posterior has sufficient capacity to represent the true posterior, and in the limit of the optimal ELBO solution. Note that the sparsity regularization term with weight $\gamma$ may introduce bias, therefore, strict consistency holds when $\gamma$ is sufficiently small, such that the regularized solution remains close to the unregularized ELBO optimum.

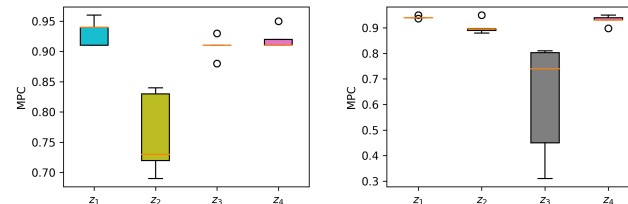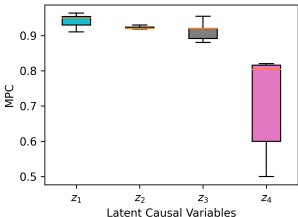

Figure 2: Performance of the proposed method under scenarios where condition (iv) is not satisfied regarding the causal influence of $z_1 \to z_2$ (consequently, $z_2 \to z_3$, and $z_3 \to z_4$). The results are in agreement with partial identifiability in Theorem 3.4, i.e., roughly speaking, latent variables that satisfy Condition (iv) are identifiable, while those that do not are not identifiable.

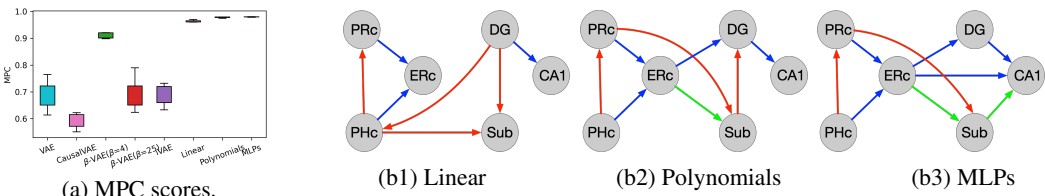

(a) MPC scores.    (b1) Linear    (b2) Polynomials    (b3) MLPs

Figure 3: (a) MPC scores achieved by different methods. Notably, the proposed MLPs achieve an outstanding average MPC of 0.981, outperforming polynomials (0.977) and linear models (0.965). (b) Recovered latent causal structures using (b1) latent linear models, (b2) latent polynomials, and (b3) latent MLPs. Results for linear models and polynomials are sourced from (Liu et al., 2024). Blue edges align with known anatomical connectivity, red edges violate anatomical constraints, and green edges are reversed directions.

**Semi-Synthetic fMRI Data**   Building on the works in Liu et al. (2022; 2024), we extended the application of the proposed method to the fMRI hippocampus dataset (Laumann & Poldrack, 2015). This dataset comprises signals from six distinct brain regions: perirhinal cortex (PRC), parahippocampal cortex (PHC), entorhinal cortex (ERC), subiculum (Sub), CA1, and CA3/Dentate Gyrus (DG). These signals, recorded during resting states, span 84 consecutive days from a single individual. Each day's data contributes to an 84-dimensional vector, e.g., $\mathbf{u}$. Our focus is on uncovering latent causal variables, therefore, we treat these six brain signals as such. Specifically, we assume that they undergo a random nonlinear mapping into the observable space, after which suitable methods can be applied to recover them.

Figure 3 presents the comparative results yielded by the proposed method alongside various other methods. Notably, the VAE, $\beta$-VAE, and iVAE models presume the independence of latent variables, rendering them incapable of discerning the underlying latent causal structure. Conversely, other methods, including latent linear models, latent polynomials, and latent MLPs, are able to accurately recover the latent causal structure with guarantees. We also include CausalVAE Yang et al. (2021) as a baseline. Among these, the MLP models outperform the others in terms of MPC. In the study by Liu et al. (2024), it is noted that linear relationships among the examined signals tend to be more prominent than nonlinear ones. This observation might lead to the presumption that linear models would be effective. However, this is not necessarily the case, as these models can still yield suboptimal outcomes. In contrast, MLPs demonstrate superior performance in term of MPC, particularly when compared to polynomial models, which are prone to instability and exponential growth issues. The effectiveness of MLPs is further underscored by their impressive average MPC score of 0.981. It is important to emphasize that while the improvement in MPC over the proposed method (also achieving 0.981) may appear marginal, compared to prior methods such as linear models (MPC 0.965) and polynomial models (MPC 0.977), this seemingly "slight" gain in MPC corresponds to a substantial difference in the recovered graph structures, which is visually illustrated in Figure 3 (b3). Moreover, we found that CausalVAE consistently produces fully connected graphs across all different random seeds, resulting in an SHD of $9.0 \pm 0.0$. MLP-based model achieves an SHD of $4.75 \pm 0.22$, outperforming the polynomial model ($5.5 \pm 0.25$) and the linear model ($5.0 \pm 0.28$). Here, we note that although the polynomial model may underperform the linear model on average, in this particular example its ability to capture non-linear relationships allows it to achieve a lower SHD.

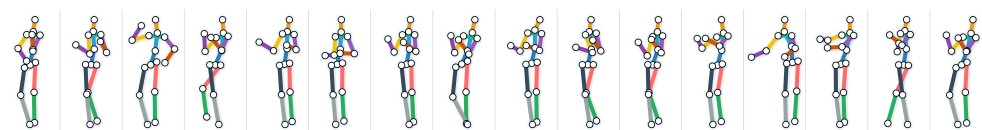

Figure 4: Some sample examples of the data we used.

**Real Human Motion Dataset**   In the final experiment, we apply the proposed method to a potential real-world application: human motion analysis. The human nervous system adapts flexibly to different motor tasks (e.g., walking, running, lifting, or fine hand manipulation). In this setting, the observed data, i.e., human skeleton poses captured via motion capture (i.e., $\mathbf{x}$), is influenced by different motor tasks $\mathbf{u}$ (Cappellini et al., 2006; Yuan et al., 2015). It is natural to consider that human motion is governed by a set of underlying latent variables (i.e., $\mathbf{z}$) (Schmidt et al., 2018; Svoboda & Li, 2018; Taylor et al., 2006; Gallego et al., 2017). Each latent variable $z_i$ capture patterns analogous to motor neuron activation dynamics (Gallego et al., 2017). The interactions among these latent variables capture the coordination dynamics between control modules involved in executing the task. Crucially, previous studies demonstrate that these interactions are task-dependent and reconfigurable (i.e., $g_i^{\mathbf{u}}$) (Doyon & Benali, 2005; Bizzi et al., 2008; d'Avella et al., 2003; Rehme et al., 2013).

For example, compared to gait velocity in the single-task condition, the networks associated with gait velocity in the dual-task condition were associated with greater functional connectivity in supplementary motor and prefrontal regions (Yuan et al., 2015). Dynamic causal modeling was applied for neuronal states of the regions of interest the motor task time series to estimate endogenous and context-dependent effective connectivity (Rehme et al., 2013). Therefore, modeling human motion with latent variables whose interaction strengths vary across motor tasks offers a biologically plausible framework for capturing the flexibility and task-specific nature of human motor control.

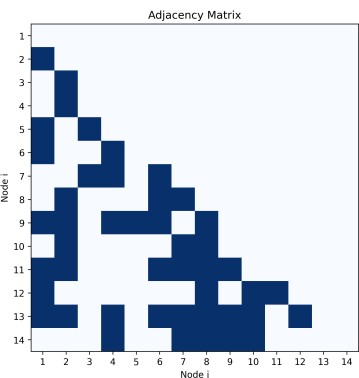

Figure 5: The estimated adjacency matrix by the proposed method.

We use the Human3.6M dataset, which provides a diverse set of 17 motion tasks such as discussion, smoking, taking photos, and talking on the phone. Following pre-processing steps from prior works (see Appendix M.1), we construct a filtered subset comprising 7 subjects and 15 motion tasks, resulting in a total of 105 distinct values of the condition variable $\mathbf{u}$. For each condition, we obtain 1,040 samples. Each sample is represented as a $2 \times 16$ matrix, where 2 denotes the spatial coordinates, and 16 corresponds to skeletal keypoints, such as joints of the head, shoulders, elbows, and knees. Figure 4 shows some samples we used in experiments after preprocessing.

Due to the absence of ground-truth semantics for latent variables, we evaluate the proposed method by intervening on each latent variable and observing the corresponding changes in the output data. This allows us to infer the potential semantic meaning associated with each latent factor. We emphasize that precisely disentangling latent variables remains a challenging task, even in synthetic settings. Therefore, our focus is on identifying dominant changes in the observed data. Together with the estimated adjacency matrix in Figure 5, we can observe from Figures 6 and 7 that the proposed method obtains potential latent causal relations, including

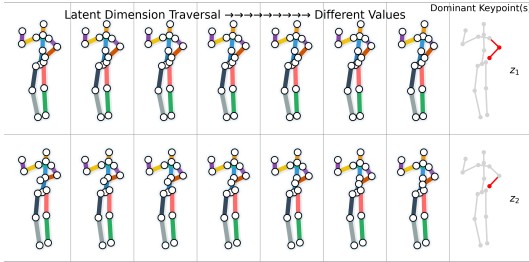

Figure 6: Intervention results for $z_1$ and $z_2$, which provide evidence of the causal relationship from the elbow to the wrist in the right hand.

from the shoulder joint to the wrist joint, and from the elbow to the wrist joint. Such results tend to be plausible, as they are closely aligned with the dynamic model of intersegmental limb interactions. For example, it has been demonstrated that the nervous system predicts and compensates for interaction torques arising from shoulder and elbow movements to adjust wrist muscle activity

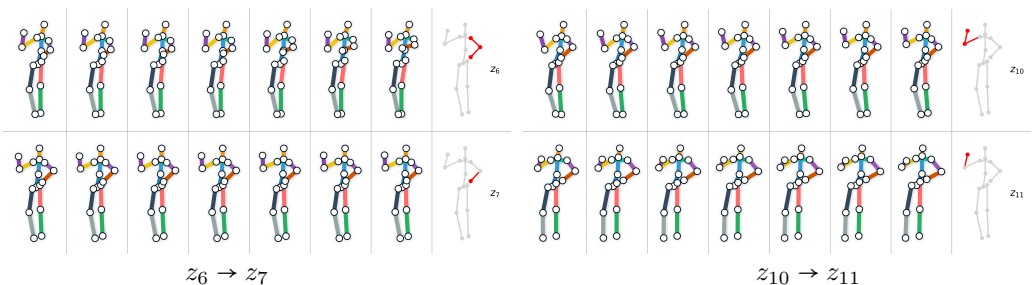

$$z_6 \rightarrow z_7 \qquad\qquad z_{10} \rightarrow z_{11}$$

Figure 7: Visualization of selected latent variable interventions. Intervention on $z_6$ leads to changes from the shoulder joint to the wrist joint, whereas intervention on $z_7$ affects only the wrist joint, indicating the causal relationship from the shoulder to the wrist. The right reveals the causal relationship from the elbow to the wrist.

reflexively, reflecting an internal model of limb dynamics (Kurtzer et al., 2008). In particular, they showed that elbow muscle activity precedes and modulates wrist muscle responses, indicating that the nervous system integrates information about elbow joint dynamics to coordinate distal muscle control effectively. See Appendix M for details.

## 6  CONCLUSION

This study makes a significant contribution by establishing a precise condition for identifying the types of distribution shifts necessary for the identifiability of latent additive noise models. We also introduce partial identifiability, applicable in cases where only a subset of distribution shifts satisfies this condition. Furthermore, we extend the results to latent post-nonlinear causal models, thereby broadening the theoretical scope. These theoretical insights are translated into a practical method, and we conduct extensive empirical testing across a wide range of datasets. Importantly, we demonstrate a promising application in learning causal representations for human motion data. We hope that this work paves the way for the development of practical methods for learning causal representations.

## 7  LIMITATIONS AND DISCUSSIONS

It should be noted that our framework relies on the Assumptions (i)-(iv), as well as the generative model defined in Eqs. (1)–(3), which are inherently untestable in practice. This limitation is not unique to our work but is a common challenge across the causal representation learning community. Precisely because of this, there is strong motivation to continue relaxing such assumptions and extending the scope of current theoretical results, which is what this work aims to do. In addition, empirical evaluations on real data provide evidence that our approach is effective, demonstrating that the insights derived from our theoretical analysis can meaningfully guide practical modeling, despite the untestable nature of the underlying assumptions.

**Ethics Statement.** This work follows the ICLR Code of Ethics. We have carefully considered potential ethical implications. No sensitive data or identifiable information is used, and we believe the societal risks are minimal.

**Reproducibility Statement.** Our experimental setup, hyperparameters, and implementation details are described in the appendix. We will release code and preprocessed data for replication after publication.

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

# Appendix

## Table of Contents

# A  RELATED WORK

Given the challenges associated with identifiability in causal representation learning, numerous existing works tackle this issue by introducing specific assumptions. We categorize these related works into three primary parts based on the nature of these assumptions.

**Special graph structure**  Some progress in achieving identifiability centers around the imposition of specific graphical structure constraints (Silva et al., 2006; Shimizu et al., 2009; Anandkumar et al., 2013; Frot et al., 2019; Cai et al., 2019; Xie et al., 2020; 2022a; Lachapelle et al., 2021). Essentially, these graph structure assumptions reduce the space of possible latent causal representations or structures, by imposing specific rules for how variables are connected in the graph. One popular special graph structure assumption is the presence of two pure children nodes for each causal variable (Xie et al., 2020; 2022a; Huang et al., 2022). Very recently, the work in (Adams et al., 2021) provides a viewpoint of sparsity to understand previous various graph structure constraints.However, any complex causal graph structures may appear in real-world scenarios, beyond the pure sparsity assumption. In contrast, our approach adopts a model-based representation for latent variables, allowing arbitrary underlying graph structures.

**Temporal Information**  The temporal constraint that the effect cannot precede the cause has been applied in causal representation learning (Yao et al., 2021; Lippe et al., 2022b; Yao et al., 2022; Lippe et al., 2022a; Li et al., 2025). The success of utilizing temporal information to identify causal representations can be attributed to its innate ability to establish causal direction through time delay. By tracking the sequence of events over time, we gain the capacity to infer latent causal variables. In contrast to these approaches, our focus lies on discovering instantaneous causal relations among latent variables.

**Changes in Causal Influences**  Recent advances have significantly developed the use of changes in causal influences within latent space as a means for identifying causal representations (Von Kügelgen et al., 2021; Liu et al., 2022; 2024; Brehmer et al., 2022; Ahuja et al., 2023; Squires et al., 2023; Buchholz et al., 2023; Varici et al., 2023; von Kügelgen et al., 2023; Ahuja et al., 2022; Varıcı et al., 2024; Varici et al., 2024). Several of these works leverage such changes in conjunction with model constraints on the mapping from latent to observed space, such as assuming linear or polynomial (Squires et al., 2023; Varici et al., 2023; Ahuja et al., 2023; Zhang et al., 2023; Varıcı et al., 2024; Jin & Syrgkanis, 2024). In contrast, our approach allows for flexibility by employing MLPs for this mapping. In addition, some works focus on imposing model assumptions on the latent causal variables, such as linear Gaussian models (Buchholz et al., 2023; Liu et al., 2022), linear additive noise models (Squires et al., 2023; Chen et al., 2024), and polynomial additive noise models (Liu et al., 2024). In contrast, our work considers both additive noise models and the more general post-nonlinear models, thereby broadening the scope of identifiable causal structures. Furthermore, prior work often relies on paired data before and after random, unknown interventions (Ahuja et al., 2022; Brehmer et al., 2022), a requirement that has been relaxed in recent work (Varici et al., 2025) to using data from two uncoupled intervention environments, whereas our method operates on fully unpaired data, a more realistic setting for applications such as biology (Squires et al., 2023; Stark et al., 2020). Additionally, some approaches require single-node interventions (von Kügelgen et al., 2023; Buchholz et al., 2023), whereas our framework allows interventions on one node while other nodes may also be simultaneously affected. In contrast to Ng et al. (2025), our framework imposes exponential-family assumptions on the latent noise variables and achieves identifiability with fewer environments in certain settings, whereas theirs relies on nonparametric assumptions and typically requires a larger number of diverse environments.

## B    DEFINITION AND LEMMAS

For ease of exposition in the following sections, we first introduce the following definition and lemmas.

**Definition B.1.** [Recursive Structural Mapping $\mathbf{h^u}$] Assume the latent causal variables follow the causal ordering $z_1 \prec z_2 \prec \cdots \prec z_\ell$ (See justification in Sec. O). Together with additive noise model assumption in Eq. 2, we have:

$$z_j = g_j^{\mathbf{u}}(z_{1:j-1}) + n_j, \qquad j = 1, \ldots, \ell. \tag{9}$$

Then the mapping $\mathbf{h^u} : \mathbf{n} \mapsto \mathbf{z}$ induced by these equations is defined recursively by

$$h_1^{\mathbf{u}}(n_1) := n_1, \tag{10}$$

and, for each $j \geq 2$,

$$h_j^{\mathbf{u}}(n_{1:j}) := g_j^{\mathbf{u}}\big(h_1^{\mathbf{u}}(n_1), h_2^{\mathbf{u}}(n_{1:2}), \ldots, h_{j-1}^{\mathbf{u}}(n_{1:j-1})\big) + n_j. \tag{11}$$

The overall mapping is

$$\mathbf{h^u}(\mathbf{n}) := \big[h_1^{\mathbf{u}}(n_1), h_2^{\mathbf{u}}(n_{1:2}), \ldots, h_\ell^{\mathbf{u}}(n_{1:\ell})\big]. \tag{12}$$

**Lemma B.1.** *The mapping, between the latent causal variables $\mathbf{z}$ and the recovered latent variables $\hat{\mathbf{z}}$ by matching the true marginal data distribution $p(\mathbf{x} \mid \mathbf{u})$, does not depend on $\mathbf{u}$.*

*Proof.* Consider Eq. (3) and assume that the function $\mathbf{f}$ is smooth and invertible, as stated in Assumption (ii). Suppose there exists an alternative solution such that $\mathbf{x} = \hat{\mathbf{f}}(\hat{\mathbf{z}})$, where $\hat{\mathbf{f}}$ is also invertible. By equating the likelihoods, we obtain:

$$\hat{\mathbf{z}} = \hat{\mathbf{f}}^{-1}(\mathbf{f}(\mathbf{z}, \varepsilon)). \tag{13}$$

Since $\varepsilon$ is independent of $\mathbf{u}$ (as per assumption (i)), and both $\mathbf{f}$ and $\hat{\mathbf{f}}$ do not depend on $\mathbf{u}$, it follows that the mapping between $\mathbf{z}$ and $\hat{\mathbf{z}}$ is also invariant with $\mathbf{u}$. $\square$

**Illustrative Example for Lemma B.1.**    Consider the following structural equations:

$$z_1 = n_1,$$
$$z_2 = \lambda(\mathbf{u}) \cdot z_1 + n_2,$$
$$x_1 = z_1,$$
$$x_2 = z_2^3 + z_1,$$

Here we neglect noise $\varepsilon$ for simplify. In this example, although the latent causal variable $z_2$ explicitly depends on $\mathbf{u}$ through $\lambda(\mathbf{u})$, the observed variables $\mathbf{x} = (x_1, x_2)$ are deterministic functions of $\mathbf{z} = (z_1, z_2)$, and this functional mapping does not involve $\mathbf{u}$.

Now suppose we attempt to recover latent variables $\hat{\mathbf{z}} = (\hat{z}_1, \hat{z}_2)$ from $\mathbf{x}$. Since $x_1 = z_1$, we have $\hat{z}_1 = x_1$. Given that $x_2 = z_2^3 + z_1 = z_2^3 + x_1$, we can rearrange and solve for $z_2$ as:

$$\hat{z}_2 = (x_2 - x_1)^{1/3}.$$

Hence, the inverse mapping:

$$\hat{\mathbf{z}} = \big(x_1, (x_2 - x_1)^{1/3}\big)$$

do not dependent of $\lambda(u)$, despite the fact that $\lambda(u)$ affects the distribution of $\mathbf{z}$. Moreover, since the mapping from $\mathbf{z}$ to $\mathbf{x}$ is also independent of $\lambda(u)$, it follows that the overall mapping between $\hat{\mathbf{z}}$ and $\mathbf{z}$ does not depend on $\lambda(u)$ either.

**Lemma B.2.** *Let $\mathbf{h^u}$ denote the mapping from $\mathbf{n}$ to $\mathbf{z}$ as defined in Definition B.1. Then, $\mathbf{h}$ is invertible, and its Jacobian determinant is equal to 1, i.e., $|\det \mathbf{J_h}| = 1$.*

*Proof.* The result follows directly from the structural form of the generative process. According to Eq. (2), each variable $z_i$ depends only on its parents and the corresponding noise variable $n_i$. This allows us to recursively express each $z_i$ in terms of its ancestral noise variables and $n_i$.

According to the definition of $\mathbf{h^u}$ in Definition B.1, we have:

$$
z_1 = \underbrace{n_1}_{h_1^{\mathbf{u}}(n_1)} \; ,
$$

$$
z_2 = g_2^{\mathbf{u}}(z_1) + n_2 = \underbrace{g_2^{\mathbf{u}}(n_1) + n_2}_{h_2^{\mathbf{u}}(n_1, n_2)},
$$

$$
z_3 = \underbrace{g_3^{\mathbf{u}}(n_1, g_2^{\mathbf{u}}(n_1) + n_2) + n_3}_{h_3^{\mathbf{u}}(n_1, n_2, n_3)}, \tag{14}
$$

$$
\vdots
$$

Due to the structure of the additive noise model and the acyclicity of the underlying causal graph (DAG), the mapping $\mathbf{h^u}$ is invertible. Moreover, its Jacobian matrix is lower triangular with ones on the diagonal, which directly implies that $|\det \mathbf{J_{h^u}}| = 1$.

$\square$

**Lemma B.3.** *Under Assumption (iv) in Theorem 3.1, consider the recursive mapping defined in Eq. (14). Let $n_{i'}$ denote the latent noise variable corresponding to a parent node $z_{i'} \in \mathrm{pa}_i$ of $z_i$, where $i' < i$. Then, the partial derivative of $h_i^{\mathbf{u}}$ with respect to $n_{i'}$ vanishes at $\mathbf{u} = \mathbf{u}_i$, i.e.,*

$$
\frac{\partial h_i^{\mathbf{u}=\mathbf{u}_i}(n_1, \ldots, n_i)}{\partial n_{i'}} = 0.
$$

*Proof.* From Eq. (14), the recursive mapping is

$$
h_i^{\mathbf{u}}(n_1, \ldots, n_i) = g_i^{\mathbf{u}}(z_1, \ldots, z_{i-1}) + n_i.
$$

Applying the chain rule gives

$$
\frac{\partial h_i^{\mathbf{u}}}{\partial n_{i'}} = \sum_{z_j \in \mathrm{pa}_i} \frac{\partial g_i^{\mathbf{u}}}{\partial z_j} \cdot \frac{\partial z_j}{\partial n_{i'}}.
$$

By Assumption (iv), there exists a parameter $\mathbf{u}_i$ such that for all parent nodes $z_j \in \mathrm{pa}_i$,

$$
\frac{\partial g_i^{\mathbf{u}=\mathbf{u}_i}}{\partial z_j} = 0.
$$

Therefore, each term in the sum above vanishes, yielding

$$
\frac{\partial h_i^{\mathbf{u}=\mathbf{u}_i}}{\partial n_{i'}} = \sum_{z_j \in \mathrm{pa}_i} 0 \cdot \frac{\partial z_j}{\partial n_{i'}} = 0.
$$

This completes the proof. $\square$

## C    THE PROOF OF THEOREM 3.1

**Theorem 3.1.** *Suppose latent causal variables $\mathbf{z}$ and the observed variable $\mathbf{x}$ follow the causal generative models defined in Eqs. (1) - (3). Assume the following holds:*

- *(i)* ~~*The noise probability density function $p_\varepsilon(\varepsilon)$ does not depend on $\mathbf{u}$ and is always finite,*~~ *The set $\{\mathbf{x} \in \mathcal{X} | \varphi_\varepsilon(\mathbf{x}) = 0\}$ has measure zero (i.e., has at most countable number of elements), where $\varphi_\varepsilon$ is the characteristic function of the density $p_\varepsilon$,*

- *(ii)* *The function $\mathbf{f}$ in Eq. (3) is smooth and invertible,*

- *(iii)* *There exist $2\ell + 1$ values of $\mathbf{u}$, i.e., $\mathbf{u}_0, \mathbf{u}_1, ..., \mathbf{u}_{2\ell}$, such that the matrix*

$$\mathbf{L} = (\boldsymbol{\eta}(\mathbf{u} = \mathbf{u}_1) - \boldsymbol{\eta}(\mathbf{u} = \mathbf{u}_0), ..., \boldsymbol{\eta}(\mathbf{u} = \mathbf{u}_{2\ell}) - \boldsymbol{\eta}(\mathbf{u} = \mathbf{u}_0)) \tag{15}$$

*of size $2\ell \times 2\ell$ is invertible. Here $\boldsymbol{\eta}(\mathbf{u}) = [\eta_{i,j}(\mathbf{u})]_{i,j}$,*

- *(iv)* *The function class of $\mathrm{g}_i^{\mathbf{u}}$ satisfies the following condition: there exists $\mathbf{u}_i$, such that, for all parent nodes $z_j$ of $z_i$, $\frac{\partial \mathrm{g}_i^{\mathbf{u}=\mathbf{u}_i}(\mathrm{pa}_i)}{\partial z_j} = 0$.*

*Then each true latent variable $z_i$ is linearly related to exactly one estimated latent variable $\hat{z}_j$, as $z_i = s_j \hat{z}_j + c_i$, for some constants $s_j$ and $c_i$, where all $\hat{z}_j$ are learned by matching the true marginal data distribution $p(\mathbf{x} \mid \mathbf{u})$.*

*Proof.* The proof of Theorem 3.1 unfolds in three distinct steps. Initially, Initially, Step I shows how the nonlinear ICA identifiability result (Khemakhem et al., 2020; Sorrenson et al., 2020) holds in our context. Specifically, it confirms that each true latent noise variable $n_i$ is related to exactly one estimated latent variable $\hat{n}_j$, e.g., $n_i = A_{i,j}\hat{n}_j + c_j$, for some constant $A_{i,j}$ and $c_i$. Building on this, Step II demonstrates a linkage between the estimated latent causal variables $\hat{\mathbf{z}}$ and the true $\mathbf{z}$, formulated as $\mathbf{z} = \boldsymbol{\Phi}(\hat{\mathbf{z}})$. Finally, Step III utilizes Lemma B.3 to illustrate that the transformation $\boldsymbol{\Phi}$, introduced in Step II, essentially simplifies to a combination of permutation and scaling, articulated as $\mathbf{z} = \mathbf{P}\hat{\mathbf{z}} + \mathbf{c}$.

**Notation** Suppose we have two sets of parameters $\boldsymbol{\theta} = (\mathbf{f}, \mathbf{T}, \mathbf{h}^{\mathbf{u}}, \boldsymbol{\eta})$ (e.g., parameters for generative model) and $\hat{\boldsymbol{\theta}} = (\hat{\mathbf{f}}, \hat{\mathbf{T}}, \hat{\mathbf{h}}^{\mathbf{u}}, \hat{\boldsymbol{\eta}})$ (e.g., parameters for the estimated model) corresponding to the same conditional probabilities, i.e., $p_{(\mathbf{f},\mathbf{T},\mathbf{h}^{\mathbf{u}},\boldsymbol{\eta})}(\mathbf{x}|\mathbf{u}) = p_{(\hat{\mathbf{f}},\hat{\mathbf{T}},\hat{\mathbf{h}}^{\mathbf{u}},\hat{\boldsymbol{\eta}})}(\mathbf{x}|\mathbf{u})$ for all pairs $(\mathbf{x}, \mathbf{u})$, where $\mathbf{T}$ denotes the sufficient statistics of the latent noise variables $\mathbf{n}$, and $\mathbf{f}$ corresponds to the mapping from latent causal variables to observed variables, both are defined in the causal generative model in Eqs. (1)–(3). The mapping $\mathbf{h}^{\mathbf{u}}$, defined in Definition B.1, maps latent noise $\mathbf{n}$ to latent causal variables $\mathbf{z}$ according to the assumed causal order and additive noise structure. $\hat{\mathbf{f}}, \hat{\mathbf{T}}, \hat{\mathbf{h}}^{\mathbf{u}}, \hat{\boldsymbol{\eta}}$ correspond to the analogous mappings and parameters in the estimated model.

**Step I:** According to the Notation above, using $p_{(\mathbf{f},\mathbf{T},\mathbf{h}^{\mathbf{u}},\boldsymbol{\eta})}(\mathbf{x}|\mathbf{u}) = p_{(\hat{\mathbf{f}},\hat{\mathbf{T}},\hat{\mathbf{h}}^{\mathbf{u}},\hat{\boldsymbol{\eta}})}(\mathbf{x}|\mathbf{u})$ for all pairs $(\mathbf{x}, \mathbf{u})$, we can leverage the technique from Step I in the proof of B.2.2 in Khemakhem et al. (2020) to conclude that if the observed distributions coincide after adding noise, then the underlying noise-free distributions must also coincide. Specifically,

$$\int p_{(\mathbf{T},\boldsymbol{\eta})}(\mathbf{n}|\mathbf{u})p_{(\mathbf{f},\mathbf{h^u})}(\mathbf{x}|\mathbf{n})\mathrm{d}\mathbf{n} = \int p_{(\hat{\mathbf{T}},\hat{\boldsymbol{\eta}})}(\mathbf{z}|\mathbf{u})p_{(\hat{\mathbf{f}},\hat{\mathbf{h}}^{\mathbf{u}})}(\mathbf{x}|\mathbf{n})\mathrm{d}\mathbf{n} \quad (16)$$

$$\Rightarrow \int p_{(\mathbf{T},\boldsymbol{\eta})}(\mathbf{n}|\mathbf{u})p_{\varepsilon}(\mathbf{x} - \mathbf{f} \circ \mathbf{h^u}(\mathbf{n}))\mathrm{d}\mathbf{n} = \int p_{(\hat{\mathbf{T}},\hat{\boldsymbol{\eta}})}(\mathbf{n}|\mathbf{u})p_{\varepsilon}(\mathbf{x} - \hat{\mathbf{f}} \circ \hat{\mathbf{h}}(\mathbf{n}))\mathrm{d}\mathbf{n} \quad (17)$$

$$\Rightarrow \int p_{(\mathbf{T},\boldsymbol{\eta})}((\mathbf{f} \circ \mathbf{h^u})^{-1}(\bar{\mathbf{x}})|\mathbf{u})|\det \mathbf{J}_{(\mathbf{f} \circ \mathbf{h^u})^{-1}}(\mathbf{x})|p_{\varepsilon}(\mathbf{x} - \bar{\mathbf{x}})\mathrm{d}\bar{\mathbf{x}}$$

$$= \int p_{(\hat{\mathbf{T}},\hat{\boldsymbol{\eta}})}((\hat{\mathbf{f}} \circ \hat{\mathbf{h}}^{\mathbf{u}})^{-1}(\bar{\mathbf{x}}|\mathbf{u}))|\det \mathbf{J}_{(\hat{\mathbf{f}} \circ \hat{\mathbf{h}}^{\mathbf{u}})^{-1}}(\mathbf{x})|p_{\varepsilon}(\mathbf{x} - \bar{\mathbf{x}})\mathrm{d}\bar{\mathbf{x}} \quad (18)$$

$$\Rightarrow \int \tilde{p}_{(\mathbf{T},\boldsymbol{\eta},\mathbf{f},\mathbf{h^u},\mathbf{u})}(\bar{\mathbf{x}})p_{\varepsilon}(\mathbf{x} - \bar{\mathbf{x}})\mathrm{d}\bar{\mathbf{x}} = \int \tilde{p}_{(\hat{\mathbf{T}},\hat{\boldsymbol{\eta}},\hat{\mathbf{f}},\hat{\mathbf{h}}^{\mathbf{u}},\mathbf{u})}(\bar{\mathbf{x}})p_{\varepsilon}(\mathbf{x} - \bar{\mathbf{x}})\mathrm{d}\bar{\mathbf{x}} \quad (19)$$

$$\Rightarrow (\tilde{p}_{(\mathbf{T},\boldsymbol{\eta},\mathbf{f},\mathbf{h^u},\mathbf{u})} * p_{\varepsilon})(\mathbf{x}) = (\tilde{p}_{(\hat{\mathbf{T}},\hat{\boldsymbol{\eta}},\hat{\mathbf{f}},\hat{\mathbf{h}}^{\mathbf{u}},\mathbf{u})} * p_{\varepsilon})(\mathbf{x}) \quad (20)$$

$$\Rightarrow F[\tilde{p}_{(\mathbf{T},\boldsymbol{\eta},\mathbf{f},\mathbf{h^u},\mathbf{u})}](\omega)\varphi_{\varepsilon}(\omega) = F[\tilde{p}_{(\hat{\mathbf{T}},\hat{\boldsymbol{\eta}},\hat{\mathbf{f}},\hat{\mathbf{h}}^{\mathbf{u}},\mathbf{u})}](\omega)\varphi_{\varepsilon}(\omega) \quad (21)$$

$$\Rightarrow F[\tilde{p}_{(\mathbf{T},\boldsymbol{\eta},\mathbf{f},\mathbf{h^u},\mathbf{u})}](\omega) = F[\tilde{p}_{(\hat{\mathbf{T}},\hat{\boldsymbol{\eta}},\hat{\mathbf{f}},\hat{\mathbf{h}}^{\mathbf{u}},\mathbf{u})}](\omega) \quad (22)$$

$$\Rightarrow \tilde{p}_{(\mathbf{T},\boldsymbol{\eta},\mathbf{f},\mathbf{h^u},\mathbf{u})}(\mathbf{x}) = \tilde{p}_{(\hat{\mathbf{T}},\hat{\boldsymbol{\eta}},\hat{\mathbf{f}},\hat{\mathbf{h}}^{\mathbf{u}},\mathbf{u})}(\mathbf{x}) \quad (23)$$

where:

- in Eq. (18), We made the change of variable $\bar{\mathbf{x}} = \mathbf{f} \circ \mathbf{h^u}(\mathbf{n})$ on the left hand side, and $\bar{\mathbf{x}} = \hat{\mathbf{f}} \circ \hat{\mathbf{h}}^{\mathbf{u}}(\mathbf{n})$ on the right hand side. Note that, here $\mathbf{f}$ and $\mathbf{h^u}$ are invertible, due to assumption (ii) and Lemma B.2.

- in Eq. (19), we introduced:

$$\tilde{p}_{(\mathbf{T},\boldsymbol{\eta},\mathbf{f},\mathbf{h^u},\mathbf{u})}(\mathbf{x}) = p_{(\mathbf{T},\boldsymbol{\eta})}((\mathbf{f} \circ \mathbf{h^u})^{-1}(\mathbf{x})|\mathbf{u})|\det \mathbf{J}_{(\hat{\mathbf{f}} \circ \hat{\mathbf{h}}^{\mathbf{u}})^{-1}}(\mathbf{x})|\mathbb{1}_{\mathcal{X}}(\mathbf{x}), \quad (24)$$

on the left hand side, and similarly on the right hand side.

- in Eq. (20), we used $*$ for the convolution operator.

- in Eq. (21), we used $F[.]$ to designate the Fourier transform.

- in Eq. 22, we dropped $\varphi_{\varepsilon}$ from both sides as it is non-zero almost everywhere (by assumption ((i))).

By taking the logarithm on both sides of Eq. 23, we have:

$$\log|\det \mathbf{J}_{\mathbf{f}^{-1}}(\mathbf{x})| + \log|\det \mathbf{J}_{(\mathbf{h^u})^{-1}}(\mathbf{z})| + \log p_{(\mathbf{T},\boldsymbol{\eta})}(\mathbf{n}|\mathbf{u})$$
$$= \log|\det \mathbf{J}_{(\hat{\mathbf{f}} \circ \hat{\mathbf{h}}^{\mathbf{u}})^{-1}}(\mathbf{x})| + \log p_{(\hat{\mathbf{T}},\hat{\boldsymbol{\eta}})}(\hat{\mathbf{n}}|\mathbf{u}), \quad (25)$$

where we assume an alternative solution exists such that $\mathbf{x} = \hat{\mathbf{f}}(\hat{\mathbf{z}}) = \hat{\mathbf{f}}(\hat{\mathbf{h}}^{\mathbf{u}}(\hat{\mathbf{n}}))$. By using the exponential family as defined in Eq. (1), we have:

$$\log|\det \mathbf{J}_{\mathbf{f}^{-1}}(\mathbf{x})| + \log|\det \mathbf{J}_{(\mathbf{h^u})^{-1}}(\mathbf{z})| + \mathbf{T}^T(\mathbf{n})\boldsymbol{\eta}(\mathbf{u}) - \log \prod_i Z_i(\mathbf{u}) = \quad (26)$$

$$\log|\det \mathbf{J}_{(\hat{\mathbf{f}} \circ \hat{\mathbf{h}}^{\mathbf{u}})^{-1}}(\mathbf{x})| + \hat{\mathbf{T}}^T(\hat{\mathbf{n}})\hat{\boldsymbol{\eta}}(\mathbf{u}) - \log \prod_i \hat{Z}_i(\mathbf{u}), \quad (27)$$

By using Lemma B.2, we have: $|\det \mathbf{J}_{(\mathbf{h^u})}| = 1$. Further, since both $\mathbf{h^u}$ and $\hat{\mathbf{h}}^{\mathbf{u}}$ must to be the same function class, we also have: $|\det \mathbf{J}_{\hat{\mathbf{h}}^{-1}}| = 1$. Given the above, Eqs. (26)-(27) can be reduced to:

$$\log|\det \mathbf{J}_{\mathbf{f}^{-1}}(\mathbf{x})| + \mathbf{T}^T(\mathbf{n})\boldsymbol{\eta}(\mathbf{u}) - \log \prod_i Z_i(\mathbf{u}) =$$

$$\log|\det \mathbf{J}_{\hat{\mathbf{f}}^{-1}}(\mathbf{x})| + \hat{\mathbf{T}}^T(\hat{\mathbf{n}})\hat{\boldsymbol{\eta}}(\mathbf{u}) - \log \prod_i \hat{Z}_i(\mathbf{u}). \quad (28)$$

Then by expanding the above at points $\mathbf{u}_l$ and $\mathbf{u}_0$, then using Eq. (28) at point $\mathbf{u}_l$ subtract Eq. (28) at point $\mathbf{u}_0$, we find:

$$\langle \mathbf{T}(\mathbf{n}), \bar{\boldsymbol{\eta}}(\mathbf{u}) \rangle + \sum_i \log \frac{Z_i(\mathbf{u}_0)}{Z_i(\mathbf{u}_l)} = \langle \hat{\mathbf{T}}(\hat{\mathbf{n}}), \bar{\hat{\boldsymbol{\eta}}}(\mathbf{u}) \rangle + \sum_i \log \frac{\hat{Z}_i(\mathbf{u}_0)}{\hat{Z}_i(\mathbf{u}_l)}. \quad (29)$$

Here $\bar{\boldsymbol{\eta}}(\mathbf{u}_l) = \boldsymbol{\eta}(\mathbf{u}_l) - \boldsymbol{\eta}(\mathbf{u}_0)$. By assumption (iii), and combining the $2\ell$ expressions into a single matrix equation, we can write this in terms of $\mathbf{L}$ from assumption (iii),

$$\mathbf{L}^T \mathbf{T}(\mathbf{n}) = \hat{\mathbf{L}}^T \hat{\mathbf{T}}(\hat{\mathbf{n}}) + \mathbf{b}. \tag{30}$$

Since $\mathbf{L}^T$ is invertible, we can multiply this expression by its inverse from the left to get:

$$\mathbf{T}(\mathbf{n}) = \mathbf{A}\hat{\mathbf{T}}(\hat{\mathbf{n}}) + \mathbf{c}, \tag{31}$$

Where $\mathbf{A} = (\mathbf{L}^T)^{-1}\hat{\mathbf{L}}^T$. According to lemma 3 in (Khemakhem et al., 2020) that there exist $k$ distinct values $n_i^1$ to $n_i^k$ such that the derivative $T'(n_i^1), ..., T'(n_i^k)$ are linearly independent, and the fact that each component of $T_{i,j}$ is univariate, we can show that $\mathbf{A}$ must be full rank.

Since we assume the noise to be two-parameter exponential family members as defined in Eq. (1), Eq. (31) can be re-expressed as:

$$\begin{pmatrix} \mathbf{T}_1(\mathbf{n}) \\ \mathbf{T}_2(\mathbf{n}) \end{pmatrix} = \mathbf{A} \begin{pmatrix} \hat{\mathbf{T}}_1(\hat{\mathbf{n}}) \\ \hat{\mathbf{T}}_2(\hat{\mathbf{n}}) \end{pmatrix} + \mathbf{c}, \tag{32}$$

Then, we re-express $\mathbf{T}_2$ in term of $\mathbf{T}_1$, e.g., $T_2(n_i) = t(T_1(n_i))$ where $t$ is a nonlinear mapping. As a result, we have from Eq. (32) that: (a) $T_1(n_i)$ can be linear combination of $\hat{\mathbf{T}}_1(\hat{\mathbf{n}})$ and $\hat{\mathbf{T}}_2(\hat{\mathbf{n}})$, and (b) $t(T_1(n_i))$ can also be linear combination of $\hat{\mathbf{T}}_1(\hat{\mathbf{n}})$ and $\hat{\mathbf{T}}_2(\hat{\mathbf{n}})$. This implies the contradiction that both $T_1(n_i)$ and its nonlinear transformation $t(T_1(n_i))$ can be expressed by linear combination of $\hat{\mathbf{T}}_1(\hat{\mathbf{n}})$ and $\hat{\mathbf{T}}_2(\hat{\mathbf{n}})$. This contradiction leads to that each true latent noise variable $n_i$ is related to exactly one estimated latent variable $\hat{n}_j$ (See APPENDIX C in (Sorrenson et al., 2020) for more details), as:

$$n_i = A_{i,j}\hat{n}_j + c_i. \tag{33}$$

Note that this result holds for two-parameter Gaussian, inverse Gaussian, Beta, Gamma, and Inverse Gamma (See Table 1 in (Sorrenson et al., 2020)).

~~For simplicity, in the following we neglect the noise term $\epsilon$ in Eq (3).~~ As a result, we can express Eq. (33) in vector form as:

$$\mathbf{n} = \mathbf{P}\hat{\mathbf{n}} + \mathbf{c}, \tag{34}$$

where $\mathbf{P}$ is a permutation with scaling matrix. ~~Note that this simplification is for convenience only, the identifiability result still holds even when the noise term is included. In the case where the noise term is included, the only difference is that the definition of $\hat{\mathbf{n}}$ in Eq. (34) becomes a subset of that in Eq. (32), thus involving a slight abuse of notation.~~

**Step II:** By Lemma B.2, we can denote $\mathbf{z}$ and $\hat{\mathbf{z}}$ by:

$$\mathbf{z} = \mathbf{h}^\mathbf{u}(\mathbf{n}), \tag{35}$$

$$\hat{\mathbf{z}} = \hat{\mathbf{h}}^\mathbf{u}(\hat{\mathbf{n}}), \tag{36}$$

where $\mathbf{h}$ is defined in B.2. Replacing $\mathbf{n}$ and $\hat{\mathbf{n}}$ in Eq. (34) by Eq. (35) and Eq. (36), respectively, we have:

$$(\mathbf{h}^\mathbf{u})^{-1}(\mathbf{z}) = \mathbf{P}(\hat{\mathbf{h}}^\mathbf{u})^{-1}(\hat{\mathbf{z}}) + \mathbf{c}, \tag{37}$$

where $\mathbf{h}$ (as well as $\hat{\mathbf{h}}$) are invertible supported by Lemma B.2. We can rewrite Eq. (37) as:

$$\mathbf{z} = \mathbf{h}^\mathbf{u}(\mathbf{P}(\hat{\mathbf{h}}^\mathbf{u})^{-1}(\hat{\mathbf{z}}) + \mathbf{c}). \tag{38}$$

Denote the composition by $\mathbf{\Phi}$, we have:

$$\mathbf{z} = \mathbf{\Phi}(\hat{\mathbf{z}}). \tag{39}$$

Note that $\mathbf{\Phi}$ must also satisfy the condition of being independent of $\mathbf{u}$, as demonstrated by Lemma B.1. Consequently, $\mathbf{\Phi}$ in Eq. (39) is independent of $\mathbf{u}$.

**Step III** Next, replacing $\mathbf{z}$ and $\hat{\mathbf{z}}$ in Eq. (39) by Eqs. 34, 35, and 36:

$$\mathbf{h^u}(\mathbf{P\hat{n}} + \mathbf{c}) = \mathbf{\Phi}(\hat{\mathbf{h}}^{\mathbf{u}}(\hat{\mathbf{n}})) \tag{40}$$

By differentiating Eq. (40) with respect to $\hat{\mathbf{n}}$

$$\mathbf{J_{h^u}P} = \mathbf{J_\Phi J_{\hat{h}^u}}. \tag{41}$$

As mentioned in Lemma B.2, without loss of generality, we can assume a causal order $z_1 \prec z_2 \prec \cdots \prec z_\ell$ so that the Jacobian $\mathbf{J_{h^u}}$ is lower triangular with ones on the diagonal. Similarly, $\mathbf{J_{\hat{h}^u}}$ can be made to follow the same lower-triangular structure, consistent with the constraints of the function class. Once this causal order is fixed, the matrix $\mathbf{P}$ is diagonal, which we label as $s_{1,1}, s_{2,2}, s_{3,3}, \ldots$ for convenience.

Next, under Assumption (iv), we show that the Jacobian $\mathbf{J_\Phi}$ must reduce to the same diagonal form as $\mathbf{P}$. Consequently, the mapping $\mathbf{\Phi}$ in Eq. (39) is a component-wise linear transformation. To this end, we examine Eq. (41) entry-wise. Matrix equality requires element-wise equality, so every entry of must match.

**Elements above the diagonal of matrix $\mathbf{J_\Phi}$**  Since the product of lower-triangular matrices is itself lower-triangular, and $\mathbf{J_{h^u}}$ and $\mathbf{J_{\hat{h}^u}}$ are a lower triangular matrices while $\mathbf{P}$ is a diagonal matrix, $\mathbf{J_\Phi}$ must be a lower triangular matrix.

Then by expanding the left side of Eq. (41), we have:

$$\mathbf{J_{h^u}P} = \begin{pmatrix} s_{1,1} & 0 & 0 & \ldots \\ s_{1,1}\frac{\partial h_2^{\mathbf{u}}(n_1,n_2)}{\partial n_1} & s_{2,2} & 0 & \ldots \\ s_{1,1}\frac{\partial h_3^{\mathbf{u}}(n_1,n_2,n_3)}{\partial n_1} & s_{2,2}\frac{\partial h_3^{\mathbf{u}}(n_1,n_2,n_3)}{\partial n_2} & s_{3,3} & \ldots \\ . & . & . & \ldots \end{pmatrix}, \tag{42}$$

by expanding the right side of Eq. (41), we have:

$$\mathbf{J_\Phi J_{\hat{h}^u}} = \begin{pmatrix} J_{\mathbf{\Phi}_{1,1}} & 0 & 0 & \ldots \\ J_{\mathbf{\Phi}_{2,1}} + J_{\mathbf{\Phi}_{2,2}}\frac{\partial \hat{h}_2^{\mathbf{u}}(n_1,n_2)}{\partial n_1} & J_{\mathbf{\Phi}_{2,2}} & 0 & \ldots \\ J_{\mathbf{\Phi}_{3,1}} + \sum_{i=2}^3 J_{\mathbf{\Phi}_{3,i}}\frac{\partial \hat{h}_i^{\mathbf{u}}(n_1,...,n_i)}{\partial n_1} & J_{\mathbf{\Phi}_{3,2}} + J_{\mathbf{\Phi}_{3,3}}\frac{\partial \hat{h}_3^{\mathbf{u}}(n_1,...,n_3)}{\partial n_2} & J_{\mathbf{\Phi}_{3,3}} & \ldots \\ . & . & . & \ldots \end{pmatrix}. \tag{43}$$

**The diagonal of matrix $\mathbf{J_\Phi}$**  By comparison between Eq. (42) and Eq. (43), we have $J_{\mathbf{\Phi}_{i,i}} = s_{i,i}$.

**Elements below the diagonal of matrix $\mathbf{J_\Phi}$**  By comparison between Eq. (42) and Eq. (43), and Lemma B.3, for all $i > j$ we have $J_{\mathbf{\Phi}_{i,j}} = 0$. For example, given the fact that the equality of two matrices implies element-wise equality, by comparing the corresponding elements of the two matrices Eq. (42) and Eq. (43), e.g., we have

$$s_{2,2}\frac{\partial h_3^{\mathbf{u}}(n_1,n_2,n_3)}{\partial n_2} = J_{\mathbf{\Phi}_{3,2}} + J_{\mathbf{\Phi}_{3,3}}\frac{\partial \hat{h}_3^{\mathbf{u}}(n_1,...,n_3)}{\partial n_2}. \tag{44}$$

By Lemma B.3, under Assumption (iv), the gradient $\frac{\partial h_3^{\mathbf{u}=\mathbf{u}_i}(n_1,n_2,n_3)}{\partial n_2} = 0$.

For later use, we first show that both $h_3$ and $\hat{h}_3$ must belong to the same function class in environment $\mathbf{u}_i$. Under Assumption (iv), note that in environment $\mathbf{u}_i$, the corresponding $z_3$ has no parent contribution, and thus equals its own latent noise term $n_3$ under additive noise models, i.e., $z_3 = n_3$. Now suppose, toward a contradiction, that $\hat{h}_3$ is not in the same function class as $h_3$ in environment $\mathbf{u}_i$, i.e., it has parent nodes. In this case, the corresponding $\hat{z}_3$ would necessarily mix at least two latent noise sources (its own latent noise and the latent noise from its parent node). This directly contradicts the identifiability of $n_3$ up to linear scaling, as established in **Step I**: $z_3 = n_3$ and should be identifiable, while $\hat{z}_3$ would mix at least two latent noise sources and thus cannot correspond to $n_3$ alone. Therefore, both $h_3$ and $\hat{h}_3$ must belong to the same function class in environment $\mathbf{u}_i$.

Since both $h_3$ and $\hat{h}_3$ belong to the same function class and satisfy Assumption (iv), this constraint on the partial derivative naturally holds for both, i.e., $\frac{\partial \hat{h}_3^{\mathbf{u}=\mathbf{u}_i}(n_1,n_2,n_3)}{\partial n_2} = 0$. This is not an additional assumption, but a property of the function class itself, analogous to how specifying an exponential family for the latent noise variables constrains all noise variables within that family (as in Eq. (25)). As a result, we have $J_{\mathbf{\Phi}_{3,2}} = 0$. By similar reasoning, this argument extends to all elements $J_{\mathbf{\Phi}_{i,j}}$ with $i > j$, establishing that all entries below the diagonal of $\mathbf{J}_{\mathbf{\Phi}}$ are zero.

As a result, the Jacobian matrix $\mathbf{J}_{\mathbf{\Phi}}$ in Eq. (41) must coincide with the permutation matrix $\mathbf{P}$. This shows that the mapping $\mathbf{\Phi}$ has a constant Jacobian equal to the permutation matrix $\mathbf{P}$, and therefore the transformation in Eq. (39) reduces to the following form:

$$\mathbf{z} = \mathbf{P}\hat{\mathbf{z}} + \mathbf{c}'. \tag{45}$$

$\square$

# D  THE PROOF OF THEOREM 3.4

**Theorem 3.4.** *Suppose latent causal variables* $\mathbf{z}$ *and the observed variable* $\mathbf{x}$ *follow the causal generative models defined in Eqs.* (1) - (3)*, under the condition that the assumptions (i)-(iii) are satisfied, for each* $z_i$,

(a) *if it is a root node or condition (iv) is satisfied, then the true* $z_i$ *is related to the recovered one* $\hat{z}_j$, *obtained by matching the true marginal data distribution* $p(\mathbf{x}|\mathbf{u})$, *by the following relationship:* $z_i = s_j \hat{z}_j + c_j$, *where* $s_j$ *denotes scaling,* $c_j$ *denotes a constant,*

(b) *if condition (iv) is not satisfied, then* $z_i$ *is unidentifiable.*

*Proof.* Since the proof process in Steps I and II in Appendix C do not depend on the assumption (iv), the results in both Eq. (42) and Eq. (43) hold. Then consider the following two cases.

- In cases assumption (iv) holds true for $z_i$, by using Lemma B.3, and by comparison between Eq. (42) and Eq. (43), we have: for all $j < i$ we have $J_{\boldsymbol{\Phi}_{i,j}} = 0$, which implies that we can obtain that $z_i = s_{i,i}\hat{z}_i + c_i$.

- In cases where assumption (iv) does not hold for $z_i$, such as when we compare Eq. (42) with Eq. (43), we are unable to conclude that the $i$-th row of the Jacobian matrix $\mathbf{J}_{\boldsymbol{\Phi}}$ contains only one element. For example, consider $i = 2$, and by comparing Eq. (42) with Eq. (43), we can derive the following equation: $s_{1,1}\frac{\partial h_2^{\mathbf{u}}(n_1,n_2)}{\partial n_1} = J_{\boldsymbol{\Phi}_{2,1}} + J_{\boldsymbol{\Phi}_{2,2}}\frac{\partial \hat{h}_2^{\mathbf{u}}(n_1,n_2)}{\partial n_1}$. In this case, if assumption (iv) does not hold (thus Lemma B.3 does not hold too), then once $J_{\boldsymbol{\Phi}_{2,1}} = s_{1,1}\frac{\partial h_2^{\mathbf{u}}(n_1,n_2)}{\partial n_1} - J_{\boldsymbol{\Phi}_{2,2}}\frac{\partial \hat{h}_2^{\mathbf{u}}(n_1,n_2)}{\partial n_1}$ holds true, we can match the true marginal data distribution $p(\mathbf{x}|\mathbf{u})$. This implies that $J_{\boldsymbol{\Phi}_{2,1}}$ does not necessarily need to be zero, and thus can be nonzero. Consequently, $z_2$ can be represented as a combination of $\hat{z}_1$ and $\hat{z}_2$, resulting in unidentifiability. Note that this unidentifiability result also show that the necessity of condition (iv) for achieving complete identifiability, by the contrapositive, i.e., if $z_i$ is identifiable, then condition (iv) is satisfied.

$\square$

# E  THE PROOF OF COROLLARY 3.8

**Corollary 3.8.** *Suppose latent causal variables* $\mathbf{z}$ *and the observed variable* $\mathbf{x}$ *follow the causal generative models defined in Eqs.* 1, 5 *and* 3*. Assume that conditions (i) - (iv) in Theorem 3.1 hold, then each true latent variable* $\bar{z}_i$ *is related to exactly one estimated latent variable* $\hat{\bar{z}}_j$, *which is learned by matching the true marginal data distribution* $p(\mathbf{x}|\mathbf{u})$, *by the following relationship:* $\bar{z}_i = M_j(\hat{\bar{z}}_j) + c_j$, *where* $M_j$ *and* $c_j$ *denote a invertible nonlinear mapping and a constant, respectively.*

*Proof.* The proof can be done from the following: since in Theorem 3.1, the only constraint imposed on the function $\mathbf{f}$ is that the function $\mathbf{f}$ is invertible , as mentioned in condition (ii). Consequently, we can create a new function $\widetilde{\mathbf{f}}$ by composing $\mathbf{f}$ with function $\bar{\mathbf{g}}$, in which each component is defined by the function $\bar{g}_i$. Since $\bar{g}_i$ in invertible as defined in Eq. (5), $\widetilde{\mathbf{f}}$ remains invertible. As a result, we can utilize the proof from Appendix C to obtain that $\mathbf{z}$ can be identified up to permutation and scaling, i.e., Eq. (45) holds. Finally, given the existence of a component-wise invertible nonlinear mapping between $\bar{\mathbf{z}}$ and $\mathbf{z}$ as defined in Eq. (5), i.e.,

$$\bar{\mathbf{z}} = \bar{\mathbf{g}}(\mathbf{z}). \tag{46}$$

we can also obtain estimated $\hat{\bar{\mathbf{z}}}$ by enforcing a component-wise invertible nonlinear mapping on the recovered $\hat{\mathbf{z}}$

$$\hat{\bar{\mathbf{z}}} = \hat{\bar{\mathbf{g}}}(\hat{\mathbf{z}}). \tag{47}$$

Replacing $\mathbf{z}$ and $\hat{\mathbf{z}}$ in Eq. (45) by Eq. (46) and Eq. (47), respectively, we have

$$\bar{\mathbf{g}}^{-1}(\bar{\mathbf{z}}) = \mathbf{P}\hat{\bar{\mathbf{g}}}^{-1}(\hat{\bar{\mathbf{z}}}) + \mathbf{c}'. \qquad (48)$$

As a result, we conclude the proof. $\qquad\square$

## F THE PROOF OF COROLLARY 3.9

**Corollary 4.9.** *Suppose latent causal variables $\mathbf{z}$ and the observed variable $\mathbf{x}$ follow the causal generative models defined in Eqs. 1, 5 and 3. Under the condition that the assumptions (i)-(iii) are satisfied, for each $\bar{z}_i$, (a) if condition (iv) is satisfied, then the true latent variable $\bar{z}_i$ is related to one estimated latent variable $\hat{\bar{z}}_j$, which is learned by matching the true marginal data distribution $p(\mathbf{x}|\mathbf{u})$, by the following relationship: $\bar{z}_i = M_j(\hat{\bar{z}}_j) + c_j$, (b) if condition (iv) is not satisfied, then $\bar{z}_i$ is unidentifiable.*

*Proof.* Again, since in Theorem 3.1, the only constraint imposed on the function $\mathbf{f}$ is that the function $\mathbf{f}$ is invertible, as mentioned in condition (ii). Consequently, we can create a new function $\widetilde{\mathbf{f}}$ by composing $\mathbf{f}$ with function $\bar{\mathbf{g}}$, in which each component is defined by the function $\bar{\mathbf{g}}_i$. Since $\bar{\mathbf{g}}_i$ is invertible as defined in Eq. (5), $\widetilde{\mathbf{f}}$ remains invertible. Given the above, the results in both Eq. (42) and Eq. (43) hold. Then consider the following two cases.

- In cases where $z_i$ represents a root node or assumption (iv) holds true for $z_i$, using the proof in Appendix D we can obtain that $z_i = s_{i,i}\hat{z}_i + c_i$. Then, given the existence of a component-wise invertible nonlinear mapping between $\bar{z}_i$ and $z_i$ as defined in Eq. (5), we can proof that there is a invertible mapping between the recovered $\hat{\bar{z}}_i$ and the true $\bar{z}_i$.

- In cases where assumption (iv) does not hold for $z_i$, using the proof in Appendix D $z_i$ is unidentifiable, we can directly conclude that $\bar{z}_i$ is also unidentifiable.

$\qquad\square$

## G UNDERSTANDING ENFORCING CAUSAL ORDER IN THE INFERENCE MODEL

In the inference model, we naturally enforce a causal order $z_1 > z_2 > \cdots > z_\ell$ without requiring specific semantic information. This does not imply that we need to know the true causal order a prior. Instead, we leverage the permutation indeterminacy in latent space, as demonstrated in (Liu et al., 2022).

For instance, suppose the underlying latent causal variables correspond to properties such as the size ($z_1$) and color ($z_2$) of an object. Permutation indeterminacy implies that we cannot guarantee whether the recovered latent variable $\hat{z}_1$ represents the size or the color. This ambiguity in the latent space, however, offers an advantage: by predefining a causal order, we enforce that $\hat{z}_1$ causes $\hat{z}_2$, without explicitly specifying the semantic meaning of these variables.

Due to the identifiability guarantee, $\hat{z}_1$, as the first node in the predefined causal order, will learn the semantic information of the first node in the true underlying causal order, e.g., the size. Similarly, $\hat{z}_2$, as the second node in the predefined causal order, will be assigned to learn the semantic feature of the second node (e.g., color). As a result, we can naturally establish a causal fully-connected graph by pre-defining causal order, ensuring the estimation of a directed acyclic graph (DAG) in inference model and avoiding DAG constraints, such as those proposed by (Zheng et al., 2018).

## H DATA DETAILS

**Synthetic Data** In our experimental results using synthetic data, we utilize 50 segments, with each segment containing a sample size of 1000. Furthermore, we explore latent causal or noise variables with dimensions of 2, 3, 4, and 5, respectively. Specifically, our analysis centers around the following

structural causal model:

$$n_i :\sim \mathcal{N}(\alpha, \beta), \tag{49}$$

$$z_1 := n_1, \tag{50}$$

$$z_2 := \lambda_{1,2}(\mathbf{u})\sin(z_1) + n_2, \tag{51}$$

$$z_3 := \lambda_{2,3}(\mathbf{u})\cos(z_2) + n_3, \tag{52}$$

$$z_4 := \lambda_{3,4}(\mathbf{u})\log(z_3^2) + n_4, \tag{53}$$

$$z_5 := \lambda_{3,5}(\mathbf{u})\exp(\sin(z_3^2)) + n_5. \tag{54}$$

In this context, both $\alpha$ and $\beta$ for Gaussian noise are drawn from uniform distributions within the ranges of $[-2.0, 2.0]$ and $[0.1, 3.0]$, respectively. The values of $\lambda_{i,j}(\mathbf{u})$ are sampled from a uniform distribution spanning $[-2.0, -0.1] \cup [0.1, 2.0]$. After sampling the latent variables, we use a random three-layer feedforward neural network as the mixing function, as described in (Hyvarinen & Morioka, 2016; Hyvarinen et al., 2019; Khemakhem et al., 2020).

**Synthetic Data for Partial Identifiability**   In our experimental results, which utilized synthetic data to explore partial identifiability, we modified the Eqs (49)-(53) by

$$\dot{z}_i := z_i + z_{i-1}. \tag{55}$$

In this formulation, for each $i$, there exists a $z_{i-1}$ that remains unaffected by $\mathbf{u}$, thereby violating condition (iv).

**Image Data**   In our experimental results using image data, we consider the following latent structural causal model:

$$n_i :\sim \mathcal{N}(\alpha, \beta), \tag{56}$$

$$z_1 := n_1 \tag{57}$$

$$z_2 := \lambda_{1,2}(\mathbf{u})(\sin(z_1) + z_1) + n_2, \tag{58}$$

$$z_3 := \lambda_{2,3}(\mathbf{u})(\cos(z_2) + z_2) + n_3, \tag{59}$$

where both $\alpha$ and $\beta$ for Gaussian noise are drawn from uniform distributions within the ranges of $[-2.0, 2.0]$ and $[0.1, 3.0]$, respectively. The values of $\lambda_{i,j}(\mathbf{u})$ are sampled from a uniform distribution spanning $[-2.0, -0.1] \cup [0.1, 2.0]$.

# I  LATENT CAUSAL GRAPH STRUCTURE

Our identifiability result, as presented in Theorem 3.1, establishes the identifiability of latent causal variables, thereby ensuring the unique recovery of the corresponding latent causal graph. This result builds upon the intrinsic identifiability of nonlinear additive noise models, as demonstrated in prior work (Hoyer et al., 2008; Peters et al., 2014), and holds regardless of any scaling applied to $\mathbf{z}$. Moreover, while linear Gaussian models are unidentifiable in a single environment (Shimizu et al., 2006), identifiability can be achieved in multiple environments (e.g., across different values of $\mathbf{u}$), supported by the principle of independent causal mechanisms (Huang et al., 2020; Ghassami et al., 2018; Liu et al., 2022).

# J  IMPLEMENTATION FRAMEWORK

We perform all experiments using the GPU RTX 4090, equipped with 32 GB of memory. Figure 8 illustrates our proposed method for learning latent nonlinear models with additive Gaussian noise. In our experiments with synthetic and fMRI data, we implemented the encoder, decoder, and MLPs using three-layer fully connected networks, complemented by Leaky-ReLU activation functions. For optimization, the Adam optimizer was employed with a learning rate of 0.001. In the case of image data experiments, the prior model also utilized a three-layer fully connected network with Leaky-ReLU activation functions. The encoder and decoder designs were adopted from (Liu et al., 2024) and are detailed in Table 1 and Table 2, respectively.

| Layer | Output / Activation |
|---|---|
| Conv2d(3, 32, 4, stride=2, padding=1) | Leaky-ReLU |
| Conv2d(32, 32, 4, stride=2, padding=1) | Leaky-ReLU |
| Conv2d(32, 32, 4, stride=2, padding=1) | Leaky-ReLU |
| Conv2d(32, 32, 4, stride=2, padding=1) | Leaky-ReLU |
| Linear(32×32×4 + size($\mathbf{u}$), 30) | Leaky-ReLU |
| Linear(30, 30) | Leaky-ReLU |
| Linear(30, 3*2) | - |

Table 1: Encoder for the image data.

| Layer | Output / Activation |
|---|---|
| Linear(3, 30) | Leaky-ReLU |
| Linear(30, 30) | Leaky-ReLU |
| Linear(30, 32×32×4) | Leaky-ReLU |
| ConvTranspose2d(32, 32, 4, stride=2, padding=1) | Leaky-ReLU |
| ConvTranspose2d(32, 32, 4, stride=2, padding=1) | Leaky-ReLU |
| ConvTranspose2d(32, 32, 4, stride=2, padding=1) | Leaky-ReLU |
| ConvTranspose2d(32, 3, 4, stride=2, padding=1) | - |

Table 2: Decoder for the image data.

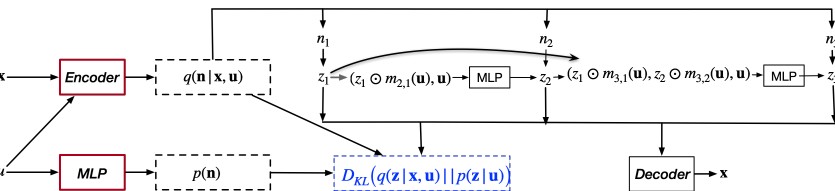

Figure 8: Implementation Framework to learn latent nonlinear models (i.e., MLP) with Gaussian noise. In this example, we demonstrate the method using 3 latent variables, however, our approach is versatile and can be effectively generalized to accommodate much larger graphs.

# K    RESULTS ON SYNTHETIC HIGH-DIMENSION DATA

In this section, we present additional experimental results on synthetic data to evaluate the effectiveness of the proposed method in scenarios with a large number of latent variables. The performance in these cases is shown in Figure 9. Compared to the polynomial-based approach in (Liu et al., 2024), the proposed method, such as MLP, achieves significantly better MCC scores, demonstrating its advantages over polynomials. This superiority becomes particularly evident as the number of latent variables increases. MLPs, being highly flexible, can effectively adapt to the growing complexity. In contrast, when the number of latent variables increases, the number of parent nodes also tends to grow, requiring polynomial-based approaches to incorporate additional nonlinear components to capture the complex relationships among latent variables, which becomes increasingly challenging.

While much of the current work on causal representation learning focuses on foundational identifiability theory, optimization challenges in the latent space remain underexplored. We hope this work not only provides a general theoretical result but also inspires further research on inference methods in the latent space.

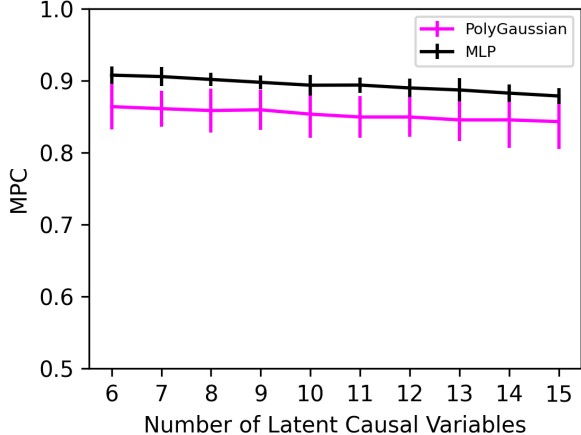

Figure 9: Performances of the proposed method on a large number of latent variables.

## L  EXPERIMENTS ON HIGH-DIMENSIONAL SYNTHETIC IMAGE DATA

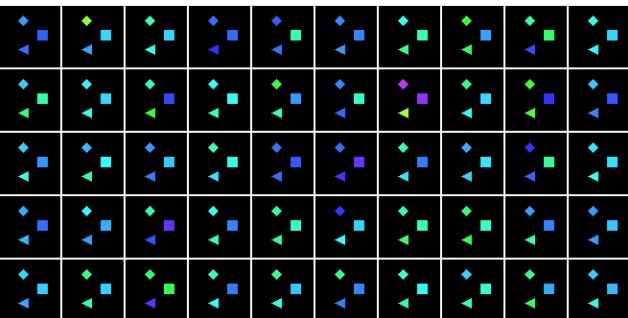

Figure 10: Samples generated by using a modified version of the chemistry dataset originally presented in (Ke et al., 2021). In this adaptation, the objects' colors (representing different states) change in accordance with a specified causal graph, e.g., 'diamond' causes 'triangle', and 'triangle' causes 'square'.

We further validate our proposed identifiability results and methodology using images from the chemistry dataset introduced by (Ke et al., 2021). This dataset is representative of chemical reactions where the state of one element can influence the state of another. The images feature multiple objects with fixed positions, but their colors, representing different states, change according to a predefined causal graph. To align with our theoretical framework, we employ a nonlinear model with additive Gaussian noise for generating latent variables that correspond to the colors of these objects. The established latent causal graph within this context indicates that the 'diamond' object (denoted as $z_1$) influences the 'triangle' ($z_2$), which in turn affects the 'square' ($z_3$). Figure 10 provides a visual representation of these observational images, illustrating the causal relationships in a tangible format.

Figure 12 presents MPC outcomes as derived from various methods. Among these, the proposed method demonstrates superior performance. In addition, both the proposed method (MLPs) and Polynomials can accurately learn the causal graph with guarantee. However, Polynomial encounters issues such as numerical instability and exponential growth in terms, which compromises its performance in MPC, as seen in Figure 12. This superiority of MLPs is further evidenced in the intervention results, as depicted in Figure 11, compared with results of Polynomial shown in Figure 13. Additional traversal results concerning the learned latent variables from other methodologies are detailed in Figure 14 (VAE), Figure 15 ($\beta$-VAE) and Figure 16 (iVAE). For these methods without identifiability, traversing any learned variable results in a change in color across all objects.

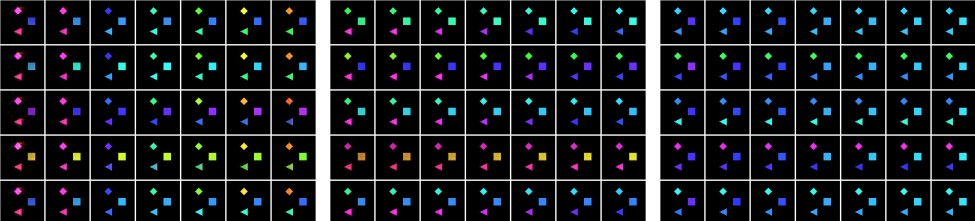

Figure 11: From left to right, the interventions are applied to the causal representations $z_1$, $z_2$, and $z_3$ learned by the proposed method (MLPs), respectively. The vertical axis represents different samples, while the horizontal axis represents the enforcement of various values on the learned causal representation.

|  | $z_1$ | $z_2$ | $z_3$ |  |  | $z_1$ | $z_2$ | $z_3$ |  |  | $z_1$ | $z_2$ | $z_3$ |
|---|---|---|---|---|---|---|---|---|---|---|---|---|---|
| $\hat{z}_1$ | 0.089 | 0.094 | 0.857 |  | $\hat{z}_1$ | 0.067 | 0.582 | 0.628 |  | $\hat{z}_1$ | 0.095 | 0.631 | 0.683 |
| $\hat{z}_2$ | 0.606 | 0.620 | 0.070 |  | $\hat{z}_2$ | 0.958 | 0.065 | 0.046 |  | $\hat{z}_2$ | 0.156 | 0.758 | 0.705 |
| $\hat{z}_3$ | 0.811 | 0.681 | 0.042 |  | $\hat{z}_3$ | 0.117 | 0.429 | 0.765 |  | $\hat{z}_3$ | 0.980 | 0.126 | 0.028 |

|  | $z_1$ | $z_2$ | $z_3$ |  |  | $z_1$ | $z_2$ | $z_3$ |
|---|---|---|---|---|---|---|---|---|
| $\hat{z}_1$ | 0.862 | 0.281 | 0.003 |  | $\hat{z}_1$ | 0.912 | 0.501 | 0.024 |
| $\hat{z}_2$ | 0.553 | 0.868 | 0.123 |  | $\hat{z}_2$ | 0.162 | 0.893 | 0.101 |
| $\hat{z}_3$ | 0.225 | 0.312 | 0.918 |  | $\hat{z}_3$ | 0.089 | 0.139 | 0.948 |

Figure 12: MPC obtained by different methods on the image dataset. From top to bottom and left to right: VAE, $\beta$-VAE, iVAE, Polynomials, and the proposed method (MLPs). The proposed method performs better than others, which is not only in line with our identifiability claims but also highlights the flexibility of MLPs.



Figure 13: From left to right, the interventions are applied to the causal representations $z_1$, $z_2$, and $z_3$ learned by Polynomials, respectively. The vertical axis represents different samples, while the horizontal axis represents the enforcement of various values on the learned causal representation.

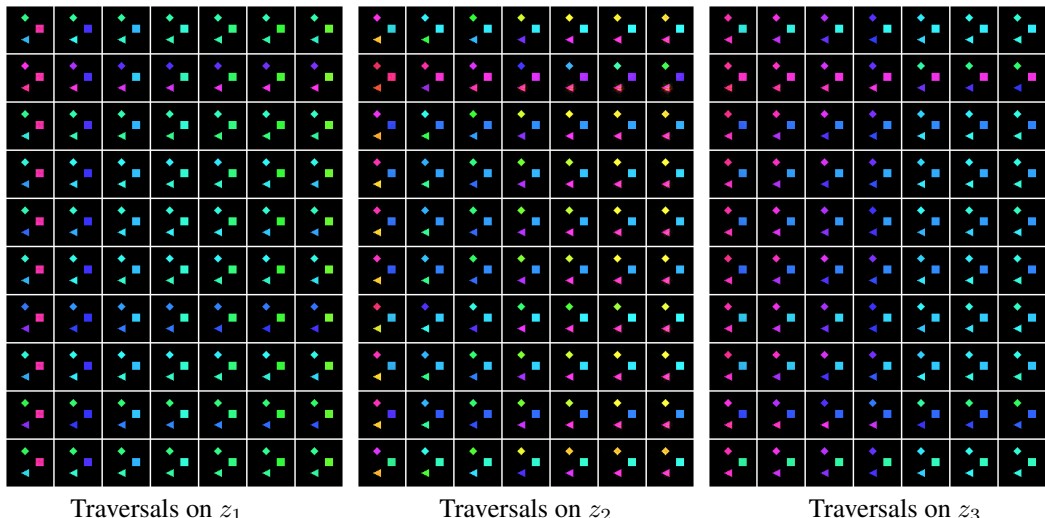

| Traversals on $z_1$ | Traversals on $z_2$ | Traversals on $z_3$ |

Figure 14: The traversal results achieved using VAE on image datasets are depicted. On this representation, the vertical axis corresponds to different data samples, while the horizontal axis illustrates the impact of varying values on the identified causal representation. According to the latent causal graph's ground truth, the 'diamond' variable (denoted as $z_1$) influences the 'triangle' variable ($z_2$), which in turn affects the 'square' variable ($z_3$). Notably, modifications in each of the learned variables lead to observable changes in the color of all depicted objects.

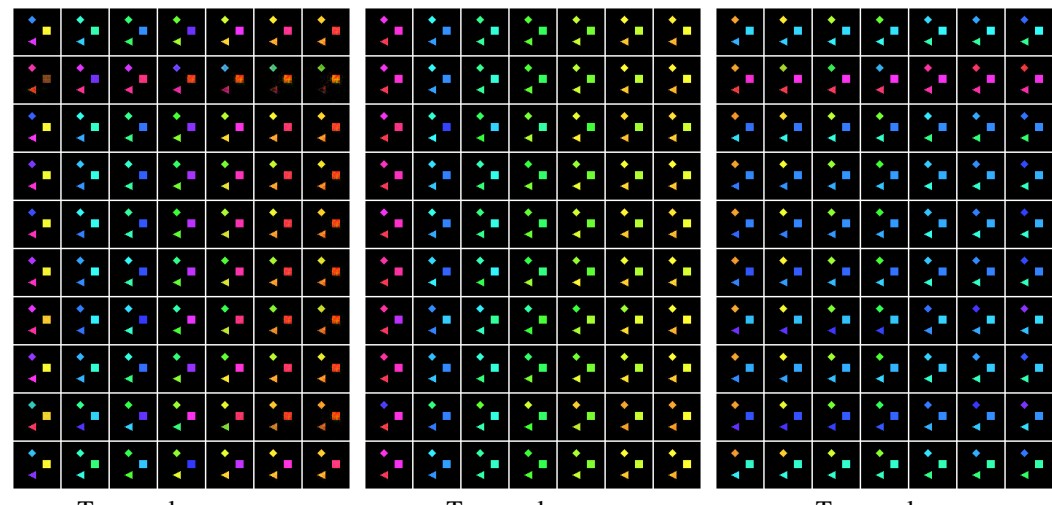

Traversals on $z_1$      Traversals on $z_2$      Traversals on $z_3$

Figure 15: The traversal results achieved using $\beta$-VAE on image datasets are depicted. On this representation, the vertical axis corresponds to different data samples, while the horizontal axis illustrates the impact of varying values on the identified causal representation. According to the latent causal graph's ground truth, the 'diamond' variable (denoted as $z_1$) influences the 'triangle' variable ($z_2$), which in turn affects the 'square' variable ($z_3$). Notably, modifications in each of the learned variables lead to observable changes in the color of all depicted objects.

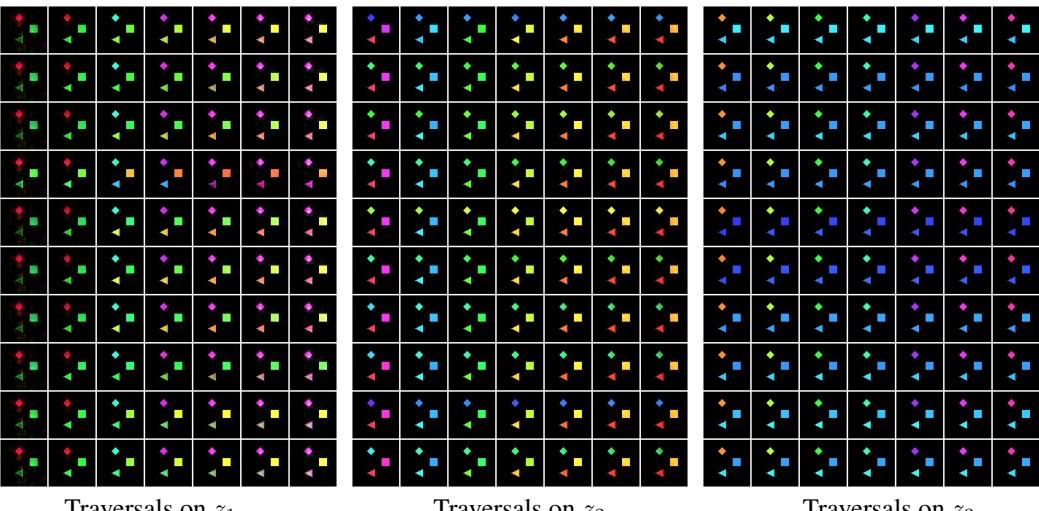

Traversals on $z_1$      Traversals on $z_2$      Traversals on $z_3$

Figure 16: The traversal results achieved using iVAE on image datasets are depicted. On this representation, the vertical axis corresponds to different data samples, while the horizontal axis illustrates the impact of varying values on the identified causal representation. According to the latent causal graph's ground truth, the 'diamond' variable (denoted as $z_1$) influences the 'triangle' variable ($z_2$), which in turn affects the 'square' variable ($z_3$). Notably, modifications in each of the learned variables lead to observable changes in the color of all depicted objects.

## M  DETAILS AND MORE RESULTS OF EXPERIMENTS ON HUMAN MOTION DATA

### M.1  PREPROCESSING

We adopt a two-stage preprocessing pipeline to construct 2D pose sequences from the Human3.6M dataset. First, we extract ground-truth 3D joint positions provided in the dataset. Each 3D pose is transformed from world coordinates into the camera coordinate frame using the associated extrinsic parameters (rotation and translation). Subsequently, we apply a perspective projection using the intrinsic parameters (focal length and principal point), and the resulting 2D coordinates are converted into image-space pixel positions based on the camera resolution. This process follows the implementation provided by https://github.com/facebookresearch/VideoPose3D/blob/main/data/prepare_data_h36m.py. Corrupted sequences (e.g., Directions for subject S11) are excluded, and only valid data is retained. The final output is stored as structured 2D keypoint arrays indexed by subject, action, and camera.

In the second stage, for each unique (subject, action) pair, we keep only the sequence from the first camera view. A unique one-hot vector is assigned to each pair, which is then concatenated to the 2D joint coordinates of every frame across all joints. To ensure balanced representation among all subject-action categories, we uniformly sample the same number of frames from each sequence based on the shortest available sequence length. This balanced and encoded dataset is then prepared for subsequent training tasks. As a result, we obtain a final dataset comprising 140 contexts $i.e.$, $\mathbf{u}$, each containing 1040 frames, with each frame represented by 2D coordinates of 16 joints.

### M.2  MORE RESULTS

In our implementation, we empirically set the number of latent variables to 14. The model is trained using the Adam optimizer with a learning rate of 1e-3 for 7000 epochs. We use the encoder designed to effectively encode 2D keypoint sequences. We employ an encoder to effectively encode 2D keypoints, where each input frame consists of $2 \times 14$ keypoint coordinates augmented with a subject-action condition vector $\mathbf{u}$. The input is first projected via a linear layer into a higher-dimensional feature space, enhancing its representational capacity. This is followed by a stack of Mixer layers, which alternate between mixing information across spatial (e.g., keypoint) and feature dimensions, thereby capturing complex dependencies both spatially and channel-wise. After all Mixer blocks, a layer normalization is applied to stabilize training. The used decoder applies multiple Mixer layers to iteratively mix spatial and channel information, followed by layer normalization for stable training. Finally, a linear layer projects the hidden features back to the keypoint coordinate dimension, producing an output that matches the original input shape of $2 \times 14$ keypoint coordinates, representing the reconstructed 2D coordinates. This decoder architecture symmetrically complements the encoder by reversing the compositional token embedding process, enabling effective recovery of keypoint positions from latent representations.

Figures 17–19 illustrate the results of intervention on each learned latent variables by our method. As discussed in the main manuscript, and supported by the estimated adjacency matrix shown in Figure 5, we observe that certain latent variables—specifically $z_1$ and $z_2$, $z_6$ and $z_7$, as well as $z_{10}$ and $z_{11}$—exhibit potential causal relationships. These include plausible dependencies such as from the shoulder to the wrist joint, and from the elbow to the wrist. Such findings are consistent with biomechanical principles of intersegmental limb dynamics.

For comparison, we also implemented the latent polynomial models proposed by (Liu et al., 2024). As shown in Figures 20–22, the learned latent representations in this baseline tend to be more entangled, lacking the interpretable structure observed in our approach.

## N  ACKNOWLEDGMENT OF LLMS USAGE

We acknowledge that large language models (LLMs) were used in this work only for word-level tasks, including correcting typos, improving grammar, and refining phrasing. No substantive content, results, or scientific interpretations were generated by LLMs. All scientific ideas, analyses, and conclusions presented in this manuscript are solely the work of the authors.

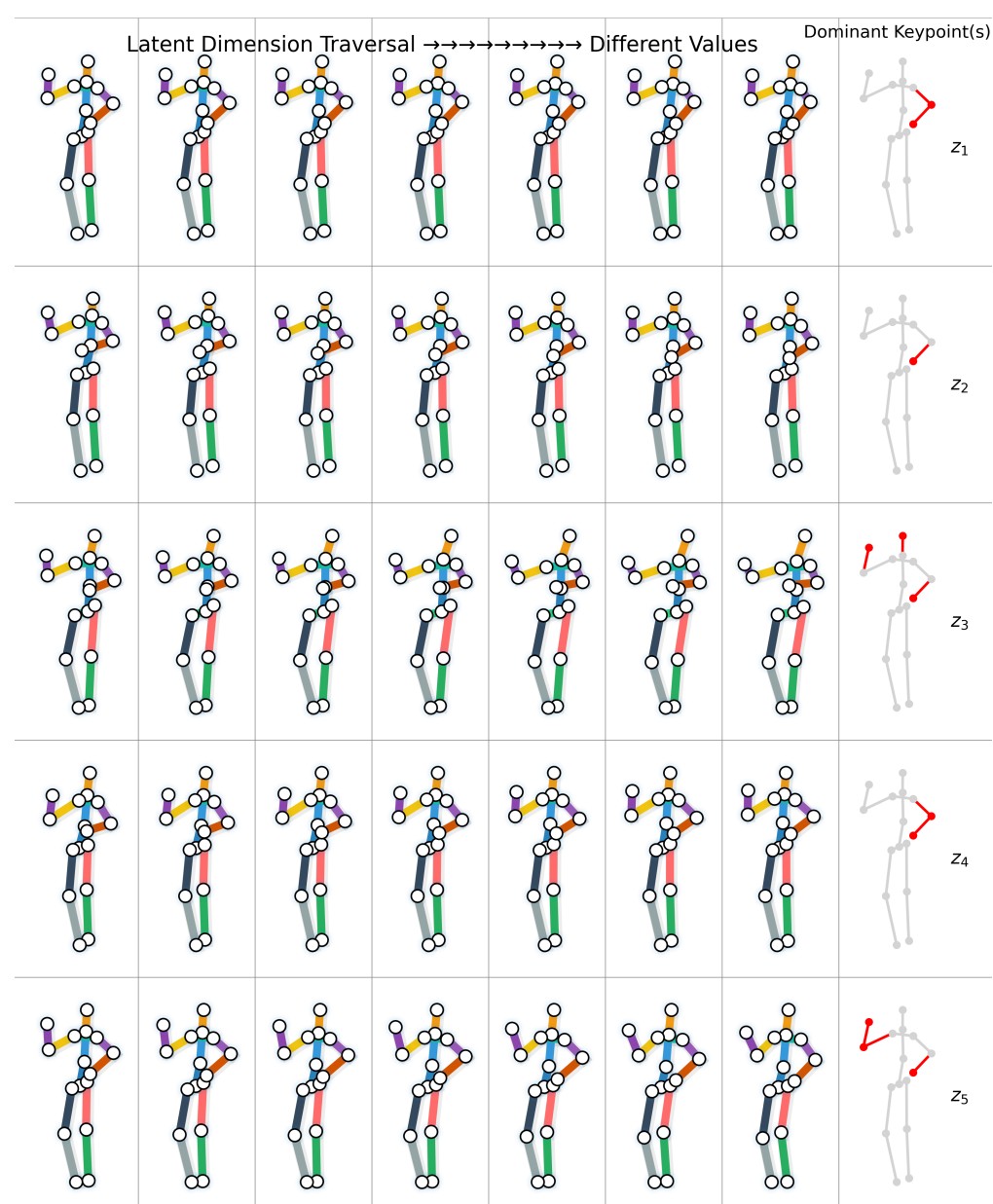

Figure 17: Complete Results of Intervention on the Estimated Latent Variables $z_1$ to $z_5$ by the Proposed Method.

## O CLARIFICATION ON CAUSAL ORDER ASSUMPTION

In the proofs (e.g., Lemma B.3 and Step III in the proof of Theorem 3.1), we assume a causal order among the latent variables $z_1 \prec z_2 \prec \cdots \prec z_\ell$. This is a *relabeling* rather than requiring known the true latent causal order.

To illustrate, consider an example with three latent variables corresponding to semantic attributes of objects: $z_{\text{color}}$ (color), $z_{\text{size}}$ (size), and $z_{\text{shape}}$ (shape), with a true causal DAG $z_{\text{color}} \to z_{\text{size}} \to z_{\text{shape}}$ (the DAG can be arbitrary). In the proof, we can always relabel the latent variables according to the coordinate indices used in the proof:

$$z_{\text{color}} \mapsto z_1, \quad z_{\text{size}} \mapsto z_2, \quad z_{\text{shape}} \mapsto z_3.$$

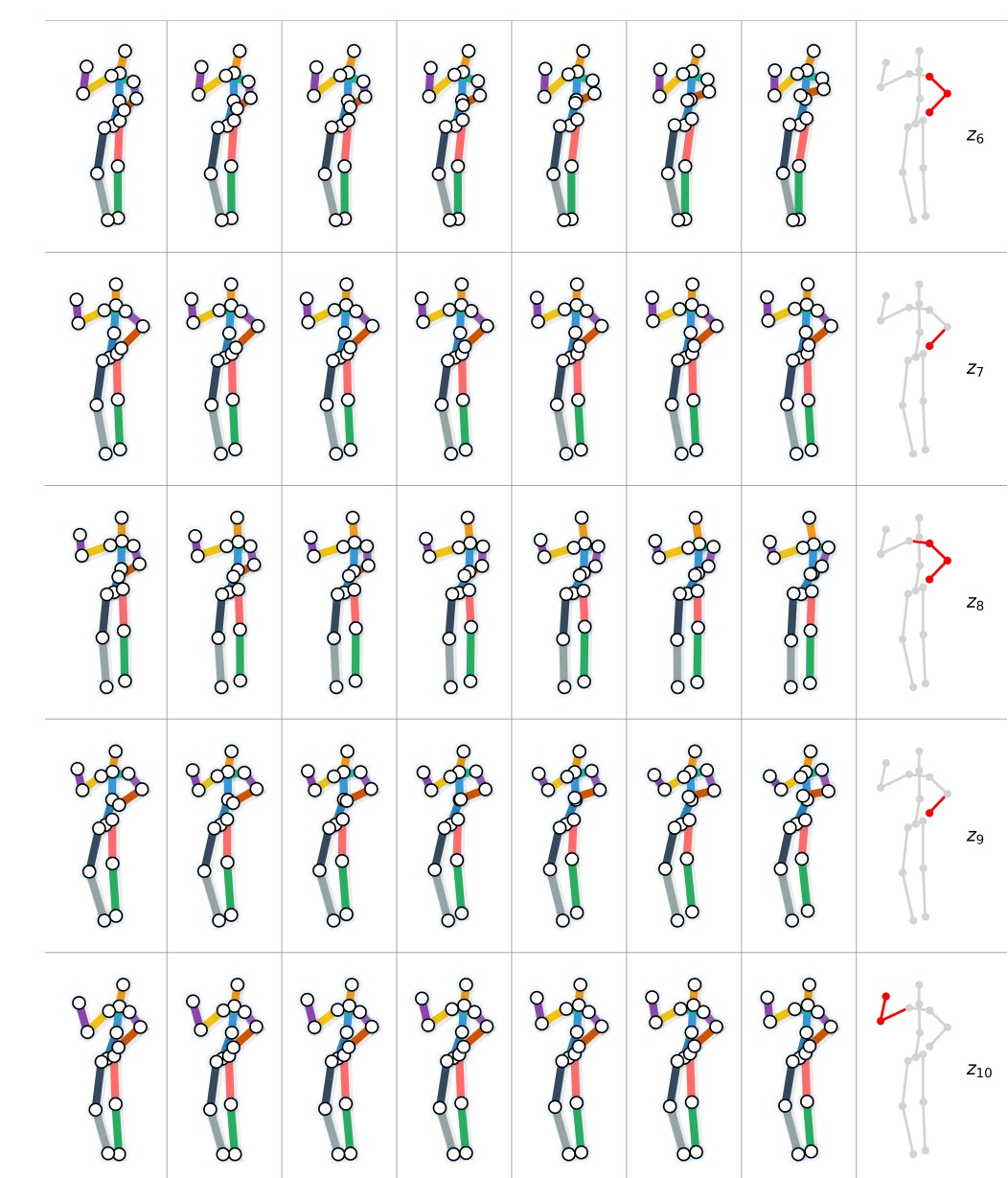

Figure 18: Complete Results of Intervention on the Estimated Latent Variables $z_6$ to $z_{10}$ by the Proposed Method.

Here, $z_1, z_2, z_3$ denote the first, second, and third nodes in the causal order of the relabeled coordinates, corresponding to color, size, and shape, respectively. *This relabeling is purely for convenience in proof and does not require knowledge of the true topological order of the latent variables.*

Thus, we claim: without loss of generality, we can consider a causal order $z_1 < z_2 < \cdots < z_\ell$, where the indices $1, 2, \ldots, \ell$ correspond to an relabeling of the latent variables. That is, the assumed causal order in the proof is purely notational and does not require knowledge of the true topological order of the original latent variables. Thus, all claims that follow from this assumed order hold without assuming access to the true causal order.

## P  RELATIONSHIP TO THE WORK OF (LIU ET AL., 2024)

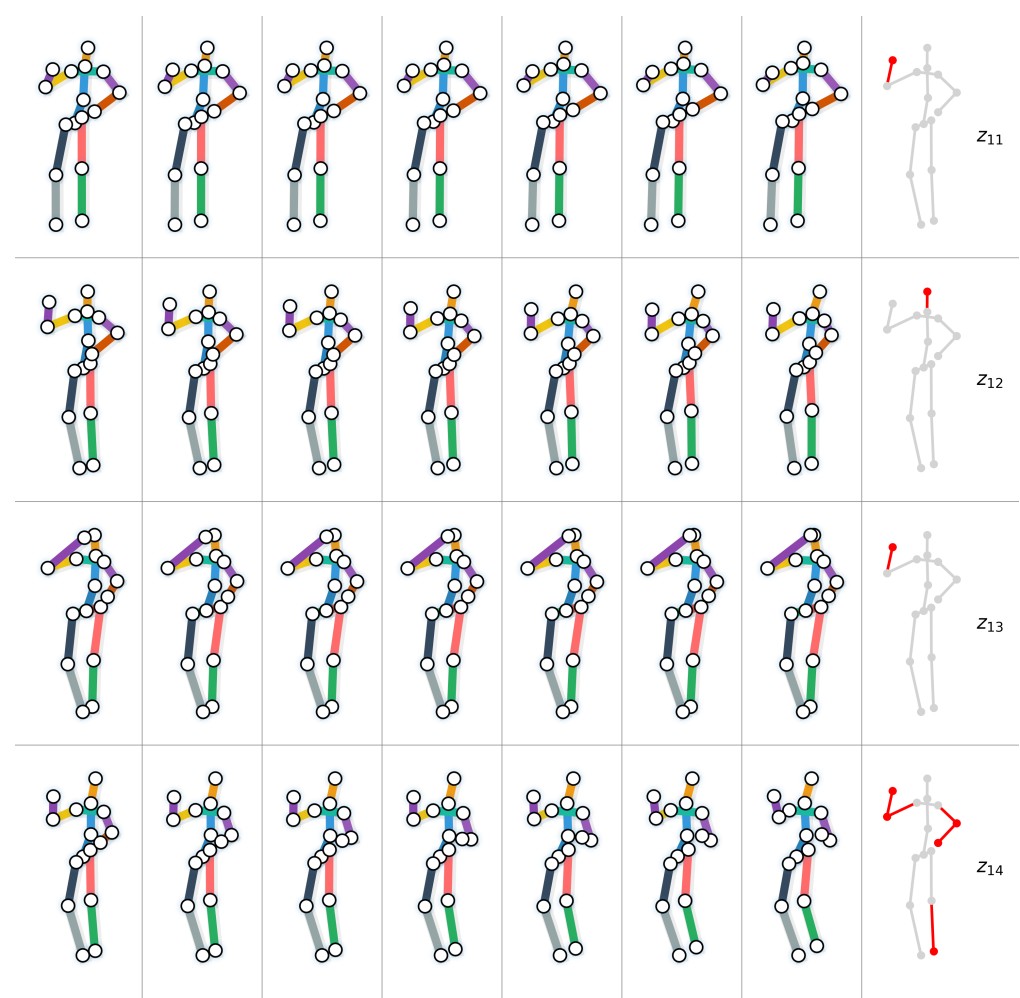

Figure 19: Complete Results of Intervention on the Estimated Latent Variables $z_{11}$ to $z_{14}$ by the Proposed Method.

We briefly clarify the relationship between our setting and the polynomial latent causal model studied in Liu et al. (2024). As already discussed in the Introduction, Related Work, and throughout our experiments, Liu et al. (2024) focuses on polynomial structural equations, whereas our work considers a more general class of additive noise models. Polynomial models may suffer from numerical instability and rapidly growing magnitudes for high-degree terms, while additive noise models avoid these issues and naturally support non-parametric instantiations (e.g., MLPs or Transformers). The polynomial setting can in fact be viewed as a special case of our formulation.

Both works build upon identifiability results from nonlinear ICA. The main conceptual difference lies in how changes in causal influences across conditions are modeled. Liu et al. (2024) specifies these changes through variations in polynomial coefficients, whereas we introduce a non-parametric conditional mechanism that captures more general forms of variation. This generalization requires substantially different proof techniques from those used in the polynomial case.

Finally, our work investigates a real-world application of causal representation learning, highlighting the practical relevance of general additive noise models beyond the polynomial setting.

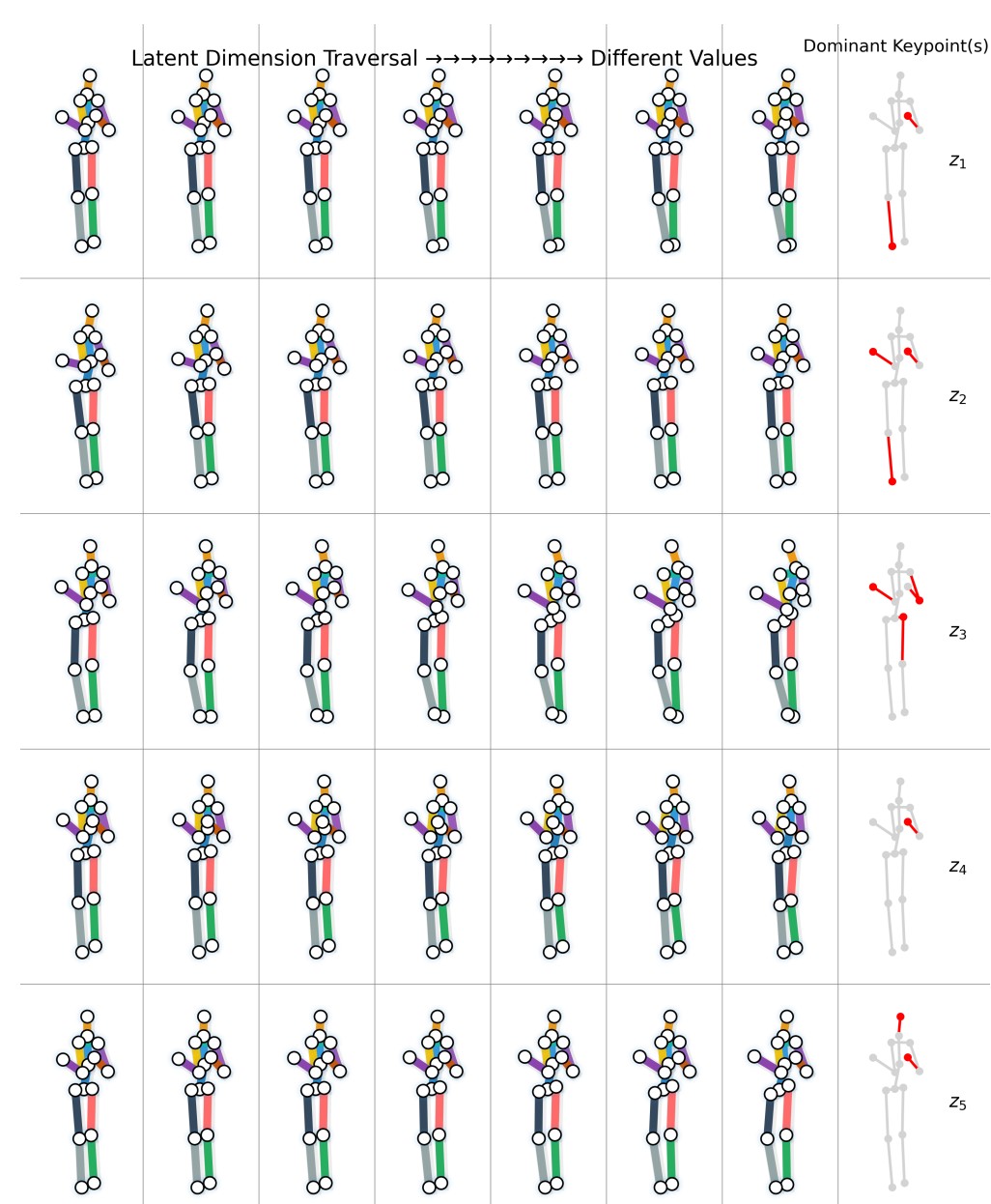

Figure 20: Complete Results of Intervention on the Estimated Latent Variables $z_1$ to $z_5$ by Latent Polynomial Model.

## Q    COMPARISON TO INTERVENTIONAL CRL

This section clarifies the distinct position of our work within the field of CRL, particularly in comparison to existing methods that rely on explicit interventional information (e.g., (Ahuja et al., 2022; Brehmer et al., 2022)).

We acknowledge that the condition formalized in Assumption (iv) is conceptually equivalent to perfect intervention at the level of the mechanism change. However, our core contribution is not in proposing the concept of intervention, but in integrating this condition into Nonlinear ICA framework, thereby achieving a significant generalization of the strict data labeling requirements of prior work.

Traditional interventional CRL methods typically leverage changes in causal mechanisms to achieve latent variable identifiability. To secure component-wise identifiability result, these methods often rely

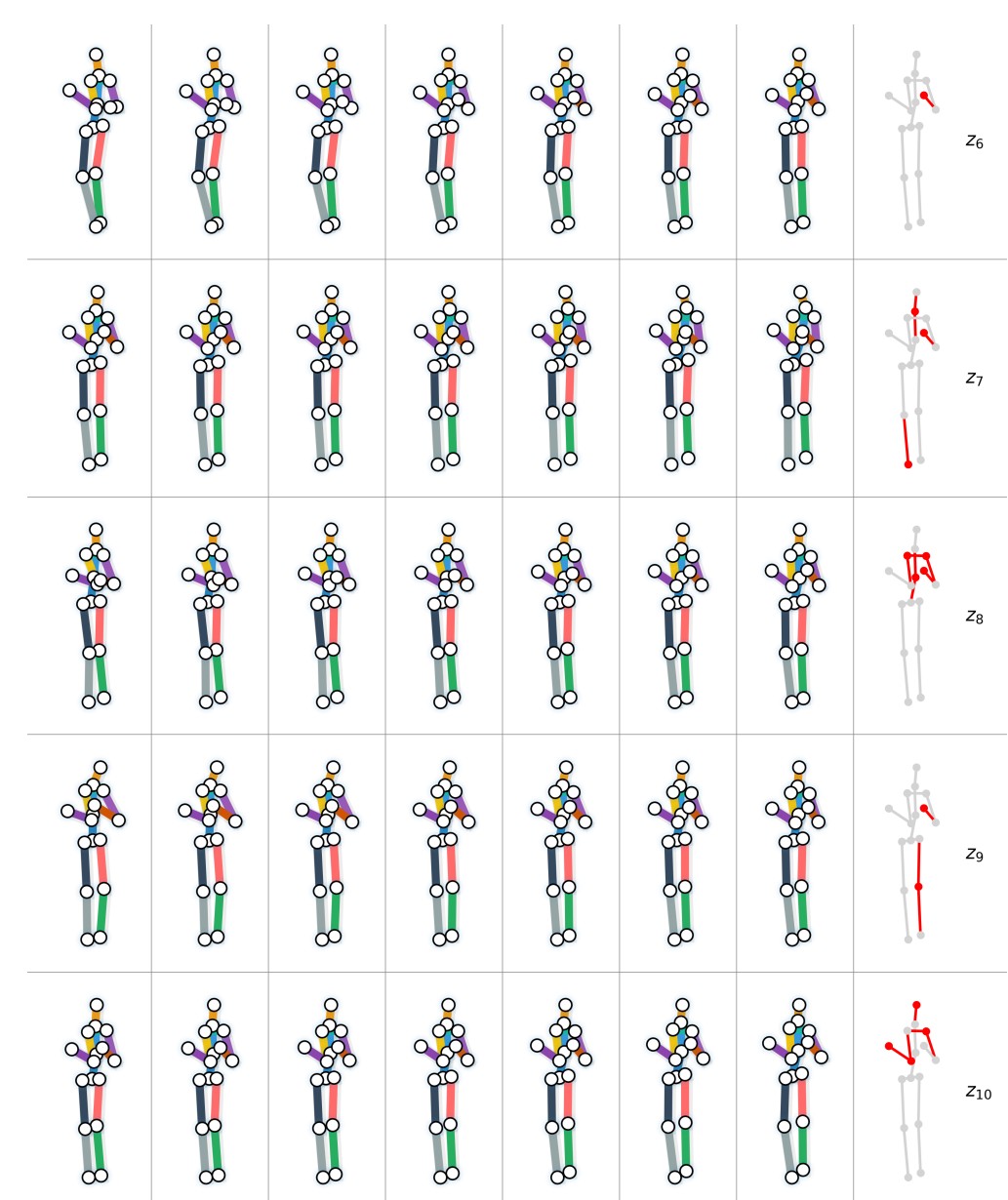

Figure 21: Complete Results of Intervention on the Estimated Latent Variables $z_6$ to $z_{10}$ by Latent Polynomial Model.

on such supervised data, characterized by the following: Many methods require prior knowledge of the specific latent node that has been intervened upon (i.e., the intervention label). Some approaches require access to paired observations (Before/After pairs) corresponding to the intervention event to isolate the "difference" signal for identifiability. This reliance on intervention labels represents a limitation for applying this research line in practical settings.

Our work successfully breaks these limitations by fusing the theoretical power of Nonlinear ICA. Consequently, our method theoretically achieves identifiability without requiring prior knowledge of which node was intervened upon or the specific semantic value of the intervention. As illustrated in Figure 23, our work resides at the intersection of two critical theoretical paradigms.

- **Path 1 (Interventional CRL):** Starting from the conceptual requirements of interventional data, we generalize the reliance on explicit labeling, mitigating data restrictions.

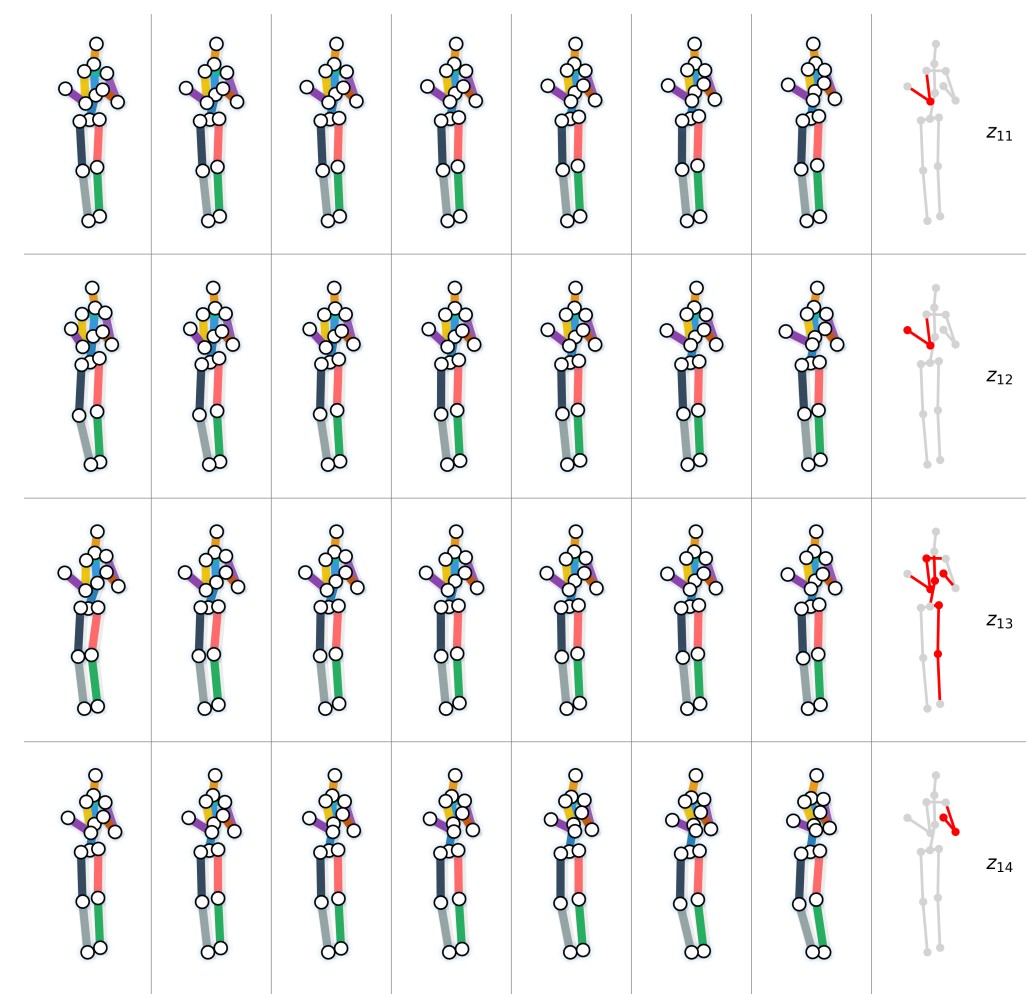

Figure 22: Complete Results of Intervention on the Estimated Latent Variables $z_{11}$ to $z_{14}$ by Latent Polynomial Model.

- **Path 2 (Nonlinear ICA):** Utilizing the statistical properties of non-stationary data, we enable the decoupling of the latent noise **n**.

These two paths converge in the proposed assumption (iv), which serves as the crucial theoretical bridge, ensuring the ultimate identifiability of the latent causal variables.

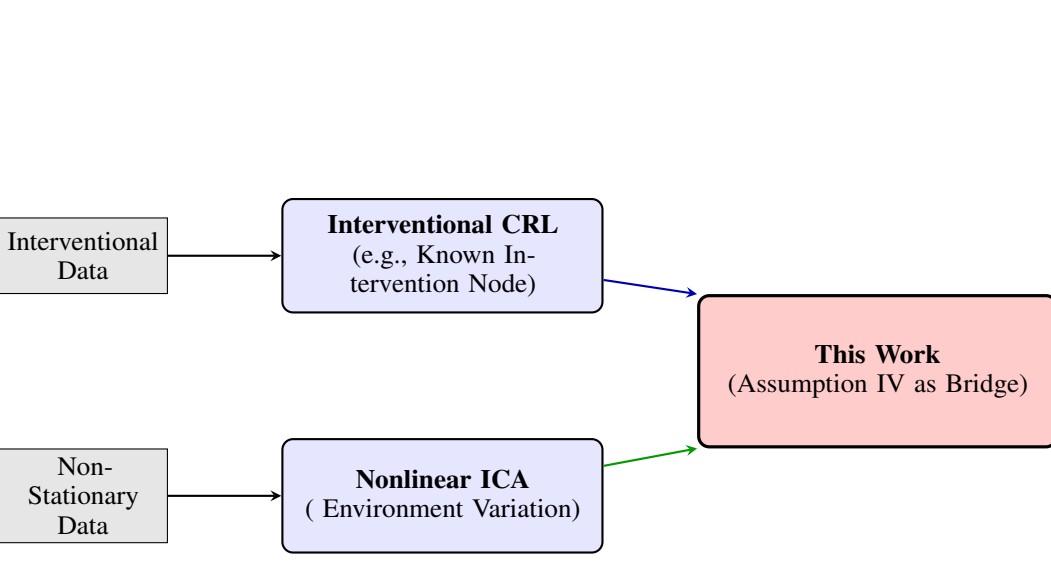

Figure 23: Comparison of previous interventional CRL and this work. Assumption (iv) serves as the crucial theoretical bridge.

