# OpenReview forum: "Towards Identifiable Latent Additive Noise Models"
_ICLR.cc/2026/Conference — Submitted to ICLR 2026_

### Official Review · Reviewer_jEPa · 2025-11-02

**Soundness:** 3
**Presentation:** 3
**Contribution:** 3
**Rating:** 8
**Confidence:** 3

**Summary:**

The paper studies identifiability for latent additive noise models (ANMs) under environment-dependent changes in causal influence among latent variables. It introduces a nonparametric “change in influence” condition (Assumption (iv)) and, combined with standard nonlinear-ICA assumptions, proves that latent ANMs are identifiable up to permutation and scaling. A VAE-style method (Sec. 4) is proposed and experiments on synthetic data, semi-synthetic fMRI, images, and Human3.6M are presented.

**Strengths:**

- Clear, general identifiability statement. Theorem 3.1 gives an if-and-only-if-style characterization (sufficiency in Thm. 3.1; necessity via Theorem 3.4(b) + Remark 3.5) for latent ANMs under nonlinear-ICA style environments. Theorem 3.4 separates nodes that satisfy (iv) (identifiable) from those that do not (unidentifiable), with an instructive invariance counter-example.

- The variational objective (Eq. 8) and sparse, u-dependent gating are a reasonable way to encourage the intended graph and environment dependence in practice, aligning with theoretical insights.

- Synthetic validation includes failure of identifiability when Assumption (iv) is violated.

**Weaknesses:**

- Requiring an environment $u_i$ where $\partial g_i^u / \partial z_j = 0$ for all parents is a perfect-intervention condition that will rarely hold exactly in real systems. The paper gives biological analogies (PPI inhibitors), but for the Human3.6M task-conditioning it’s unclear that such exact zeros occur.

- Assumptions that the mapping $f$ is injective and sufficient number of environments may be restrictive.

**Questions:**

You rely on a pre-fixed causal order in inference (App. G). How robust are results to random order permutations?

---

> ### Author Response · Authors · 2025-11-22
>
> We sincerely appreciate your time spent reviewing our draft and your positive feedback on our work.
>
> ----
>
> **Q1: Human3.6M task-conditioning it’s unclear that such exact zeros (perfect intervention) occur.**
>
> **R1:** We thank the reviewer for raising this crucial point. While it is challenging to clarify weather perfect interventions is possible or not in biological systems, neuroscience provides some evidence for task-dependent down-regulation of specific neural interactions. Studies in motor learning [1] and modular motor control [2] show that certain circuit pathways may become effectively silent, with their effective connectivity reduced to near-zero levels under specific task conditions. Thus, although biological systems rarely exhibit truly perfect interventions, these near-zero effective connections provide a realistic approximation of perfect interventions at the level of latent dynamical interactions.
>
> [1] Doyon, Julien, Virginia Penhune, and Leslie G. Ungerleider. "Distinct contribution of the cortico-striatal and cortico-cerebellar systems to motor skill learning." Neuropsychologia 41.3 (2003): 252-262.
>
> [2] Bizzi, Emilio, et al. "Combining modules for movement." Brain research reviews 57.1 (2008): 125-133.
>
>
>
> **Q2: Assumptions that the mapping $\mathbf{f}$ is injective and sufficient number of environments may be restrictive.**
>
> **R2** Thanks for pointing this out!
>
> Regarding the mapping, essentially, if one aims to exactly recover the latent variables, it is typically necessary to assume that the mapping is invertible. However, this assumption may be restrictive in real applications. In such cases, it may be possible to achieve approximate identifiability, which is an interesting direction for future exploration. To the best of our knowledge, the invertibility assumption has been adopted in most current CRL works.
>
> Furthermore, you are correct that the number of environments is another limiting factor. As in most CRL studies related to using multi-environmental data, we currently believe that the number of environments may be roughly comparable to the number of latent variables (There are some works consider special constraints on the graph structure, which may not need this). This is also one reason why partial identifiability is an interesting topic. Investigating identifiability under a limited number of environments is a promising direction for future research.
>
> **Q3: You rely on a pre-fixed causal order in inference (App. G). How robust are results to random order permutations?**
>
> **R3:** We have included a section (Sec. G) in the appendix to clarify this point. This is done without loss of generality. To illustrate, consider two latent causal variables: size ($z_1$) and color ($z_2$), with $z_1$ causing $z_2$. After recovery, the estimated variables $\hat z_1$ and $\hat z_2$ may correspond to either size ($z_1$) or color ($z_2$) due to permutation indeterminacy. This allows us to pre-define a causal order in the inference model, e.g., $\hat z_1 \to \hat z_2$, without knowing which variable represents size or color. Thanks to the identifiability guarantee, the model then learns the correct causal relationship from the data, effectively enforcing the true latent semantics (e.g., $\hat z_1$ is enforced to capture size information, and $\hat z_2$ is enforced to capture color information). Please see Sec. G for more details if you have further concerns.

---

### Official Review · Reviewer_nLhX · 2025-11-04

**Soundness:** 2
**Presentation:** 2
**Contribution:** 2
**Rating:** 2
**Confidence:** 5

**Summary:**

This paper investigates the causal representation learning problem with context or "surrogate" variables. Using prior results from nonlinear ICA literature, the authors demonstrate that variable recovery is possible for every node for which a known surrogate value exists that correspond to a perfect intervention (with recovery up to elementwise affine transformation).

**Strengths:**

Node-level or partial identifiability results are newly emerging in CRL literature, and this paper demonstrates that node-level identifiability is achievable up-to-scaling-and-shift under relatively mild conditions on the latent model using perfect/stochastic hard interventions.

**Weaknesses:**

The proof of theorem 3.1, while clever, overlooks an assumption. In order to impose the Jacobians of $\mathbf{h}\_{\mathbf{u}^i}$ and $\hat{\mathbf{h}}\_{\mathbf{u}^i}$ to have the same structure, one requires (i) $\mathbf{u}^i$ value to be known, and (ii) $i$ to be known. In interventional CRL language, this is equivalent to requiring access to a perfect intervention (corresponding to using $\mathbf{u}^i$ under assumption (iv)) on node $i$ while knowing a topological order $(1, 2, \dots )$ among the intervention targets $i \in \\{1, 2, \dots \\}$. In summary, (i) theorem statements throughout the paper require to clearly communicate this fact that "we require to know values of both $i$ and $\mathbf{u}^i$ to recover $\mathbf{z}_i$", and (ii) known latent topological order assumption -- which is used implicitly in the proofs -- must be stated explicitly in the theorem statements.

While these problems are not critical in isolation, context matters. In the case of this paper, the context is the positioning of the paper as a whole. "Known $i$ known $\mathrm{u}^i$" assumption literally is what is in other CRL contexts called interventions. Therefore, the following claim from the paper becomes strictly incorrect:

> Assumption (iv), originally introduced by this work, provides a condition that characterizes the types of change in causal influences contributing to identifiability ...
>
> ...
>
> We emphasize that assumption (iv) is our key contribution, formulating changes in causal influence as constraints on the function class and thus distinguishing our work from previous studies

Worse yet, it defeats the very novelty this paper tries to establish: generalizing the restrictiveness of existing (interventional) CRL papers in data availability. Unfortunately, these problems are inherent to the current _narrative_ of the paper.

-----

A remark on remark 3.5. The technically correct necessary condition proven in the theorem is "for a system to be identifiable, (1) it must not obey the problem setting or any of assumptions (i)--(iii), and/or (2) it must satisfy assumption (iv). This is (for obvious reasons) no more informative than the statement in theorem 3.4(b), which, while interesting, should not be presented as a general non-identifiability result.

-----

A note on the proofs. They are correct to my investigation barring the problems I raised. However, many of the arguments that are required to prove the said statements are outright missing from the current document. Take the example of step III of proof of theorem 3.1. Considering that $\mathbf{J}$ are all _functions_, it takes special effort to ensure no arguments are invalidated by making structure-based claims. Most cases are resolved since the diagonals of $\mathbf{J}\_{\mathbf{h}}$ are set to $1$, but it is not obvious to a reader. Similarly, it is essential to show that if both $\mathbf{J}\_{\mathbf{h}}$ and $\mathbf{J}\_{\hat{\mathbf{h}}}$ are (i) lower triangular and (ii) have one-hot $j$-th rows, then the $j$-th row of $\mathbf{J}\_{\mathbf{\Phi}}$ must also be one-hot. Even then, why does this imply $\hat{\mathbf{z}}_i$ is an _affine_ function of $\mathbf{z}_i$? It could be any diffeomorphism? But the math works out -- as you noted -- such that $\mathbf{J}\_{\mathbf{\Phi}}$ is a constant matrix. All these details are missing currently. I strongly suggest the authors go through these proofs and fill in all the details. This will help to prevent overlooking assumptions similar to what I stated above.

**Questions:**

I defer to weaknesses section for my comments. I do not have any questions currently to the authors.

---

> ### Author Response · Authors · 2025-11-22
>
> We thank the reviewer for the highly insightful and constructive comments. The points raised regarding the implicit assumptions in our proofs, particularly concerning the topological order and the knowledge of intervention specifics (u_i and i), are critical and have prompted us to clarify the theoretical structure and refine our narrative. We believe the reviewer's concerns stem from a misunderstanding of how we leverage the inherent properties of the Additive Noise Model (ANM) within the Nonlinear Independent Component Analysis (ICA) framework.
>
> ----
>
> **Q1: require to know values of both ui and i, and known latent topological order**
>
> **R1:** We would like to clarify two primary points:
>
> The Role of Latent Topological Order: Notational Convenience vs. Known Prior
>
> The requirement for a known latent topological order $z_1 < ... < z_{\ell}$ is purely a notational convention and a mechanism for relabeling, which is commonplace in identifiability proofs for causal models and does not sacrifice generality.
>
> Inherent DAG Structure: For any arbitrary set of latent variables $[z_1, z_2, \ldots z_\ell]$ generated by a DAG, a topological ordering $\pi: [\ell] \to [\ell]$ always exists such that $z_{\pi(1)} \prec z_{\pi(2)} \prec \dots \prec z_{\pi(\ell)}$.
>
> Relabeling for Consistency: Since a permutation of the latent variables can be absorbed into the mapping from latent to observed space, we are free to relabel these variables (In fact, this has been used in prior work, e.g., Remark 1 in [1]. We also provide Section O in the revised manuscript for a more detailed clarification). This allows us to consistently define the recursive structural mapping $\mathbf{h}^\mathbf{u}$ (Definition B.2) such that its Jacobian $\mathbf{J}_{\mathbf{h}^\mathbf{u}}$ is lower triangular with ones on the diagonal.
>
> Conclusion: This choice is made solely for mathematical convenience to ensure a consistent proof structure for coordinate $i$. It does not require us to know the true causal DAG structure beforehand, as the inferred order reflects one of the valid topological orderings. We have clarified this point in Section 5 (and Appendix P) of the revised manuscript.
>
> Knowledge of Intervention Specifics ($i$ and $\mathbf{u}_i$):
>
> Based on the above, we clarify that our method does not require such prior knowledge. In our proof, $i$ simply denotes a coordinate in the re-labeled latent variable. Since $i$ is only a positional index, its semantic identity is unknown, we do not know which latent variable it corresponds to in reality. Consequently, our method only relies on the existence of a mechanism shift (perfect intervention) at some coordinate.
>
> In summary, we ackownledge prior methods related to interventional CRL typically require knowledge of which variable is intervened and the exact intervention value to enforce constraints. In contrast, our method achieves identifiability without requiring these labels, making it more general and applicable to settings where intervention labels are unknown.
>
>
> **Q2: Therefore, the claim regarding assumption (iv) in the paper becomes strictly incorrect.**
>
> **R2:** Since this concern is based on the misunderstandings mentioned in **Q1**, based on the clarifications provided above, we believe the concerns regarding the claim of Assumption (iv) should be clearly solved.
>
>
> [1] Squires, Chandler, et al. "Linear causal disentanglement via interventions." International conference on machine learning, 2023.

---

> ### Author Response · Authors · 2025-11-22
>
> **Q3: Worse yet, it defeats the very novelty this paper tries to establish: generalizing the restrictiveness of existing (interventional) CRL papers in data availability. Unfortunately, these problems are inherent to the current narrative of the paper.**
>
>
> With the clarifications provided in **Q1**, we believe that the reviewer should now understand how our foundational approach differs from some previous interventional CRL methods, e.g., do not need known causal order, do not need to know where intervention happens, only that it happens somewhere. We now provide a high-level view to clarify how and why this work generalizes the restrictiveness of some existing CRL papers in data availability.
>
> To compare the approaches, let us consider two lines: Interventional CRL and the framework presented in this work:
>
> *Interventional CRL (Previous Work):*
>
>  In general (though not universally), this line of research relies on *before/after* pairs of observations to use that "difference" signal to identify latent variables. To enable this, it may requires explicit knowledge of: 1) the state before the change, 2) the state after the change, and 3) which variable was intervened upon to connect the two states. This requirement for specific intervention labels and paired data constitutes a current important limitation in this research line.
>
> *This Work:*
>
> Our framework is grounded in *multi-environmental data*, which has emerged as a promising direction in causal representation learning in recent years. Since such data are collected from different environments, it is generally difficult to obtain intervention labels, e.g., which latent variables have been intervened. This challenge motivates the following result: by carefully leveraging the distributional differences across environments, we can explore the possibility of achieving latent variable identifiability *without relying on intervention-specific label information*. In this context, our work investigates what assumptions on multi-environmental data are sufficient to guarantee identifiability for latent additive noise models.
>
>
>
>
> *Generalization via Two-Directional Constraints:*
>
> After identifying latent noise $\mathbf{n}$, this duality in constraints (knwon noise $\mathbf{n}$ and known observation $\mathbf{x}$) enables us to break the limitation of requiring explicit knowledge of *which exact node* the intervention happened on. Interestingly, despite their different starting points between interventional CRL and multi-environmental data, assumption (iv) plays a crucial role in building a bridge between these two research lines: both ultimately converge to the same condition, \textbf{perfect intervention}, to achieve identifiability of the causal variables.
>
> We have added Section Q in the new version for comparison to interventional CRL.
>
> We hope this clarification helps to resolve the reviewer’s concerns.
>
>
>
> **Q4: Remark 3.5 and Proof Details in Step III**
>
> Thanks for the comment, we agree your point. The statement in Remark 3.5 is not rigorous, as also pointed by Reviewer 2wYV. We have removed this remark from the manuscript. We also thank the reviewer for carefully checking Step III of our proof and confirming its correctness, while pointing out missing details that could make the argument unclear to the reader. Following your suggestion, we have fully revised and expanded Step III, providing more explicit reasoning on the structure of the Jacobians, the element-wise comparison, and the derivation of the component-wise linear form of $\mathbf{\Phi}$. We believe that these additions clarify the logical steps, address the concerns raised, and improve the overall presentation of the proof.
>
>
> We hope this response clarifies misunderstandings regarding our work, particularly concerning the assumptions of the causal order, the knowledge of intervention labels, and the differences in starting points compared to interventional CRL.
>
>
> If you have any further concerns or questions, please do not hesitate to let us know.

---

> > ### Comment · Reviewer_nLhX · 2025-11-22
> >
> > I thank the authors for their comprehensive response. I need to clarify my main concern.
> >
> > In the proof of Thm 3.1, step III, eqs. (42) (elements of $\mathbf{J}\_{\mathbf{h}^{\mathbf{u}}} \mathbf{P}$) and (43) (elements of $\mathbf{J}\_{\mathbf{\Phi}} \mathbf{J}\_{\hat{\mathbf{h}}^{\mathbf{u}}}$) in the updated paper are valid under provided assumptions.
> >
> > However, to prove that the lower triangular entries of  $\mathbf{J}\_{\mathbf{\Phi}}$ are also zero, the authors use the following argument:
> >
> > > By Lemma B.3, under Assumption (iv), the gradient $\frac{\partial h^{\mathbf{u}=\mathbf{u}^i}\_3(n\_1,n\_2,n\_3)}
> > {\partial n\_2} = 0$. Since both $h\_3$ and $\hat{h}\_3$
> > belong to the same function class and satisfy Assumption (iv), this constraint on the partial derivative naturally holds for both, i.e., $\frac{\partial \hat{h}^{\mathbf{u}=\mathbf{u}^i}\_3 (n\_1,n\_2,n\_3)}{\partial n\_2} = 0$. This is not an additional assumption, but a property of the function class itself, analogous to how specifying an exponential family for the latent noise variables constrains all noise variables within that family (as in Eq. (25)).
> >
> > If we recall assumption (iv),
> > > The function class of $g^{\mathbf{u}}\_i$ satisfies the following condition: there exists $\mathbf{u}^i$, such that, for all parent nodes $z\_j$ of $z\_i$, $\frac{∂g^{\mathbf{u=u}^i}\_i (pa\_i)}{∂z\_j} = 0$.
> >
> > The "function class" argument you use requires the assumption to be read "for all $i$, exists $\mathbf{u}^i$ such that for all $g^{\mathbf{u}}\_i$, $\mathbf{u}^i$ is a hard intervention on $z\_i$", whereas the natural and standard way to interpret it is "for all $i$, for all $g^{\mathbf{u}}\_i$, exists $\mathbf{u}^i$ such that $\mathbf{u}^i$ is a hard intervention on $z\_i$".
> >
> > If you use the standard interpretation, you cannot in general ensure that two models will match in the value of $\mathbf{u}^i$: this is completely expected if you want to assume completely unknown intervention labels
> >
> > On the other hand, the current (nonstandard) reading has the problem that we are by construction aligning the $\mathbf{u}^i$ values in all members of the function class.
> >
> > **Main point:** Aligning the $\mathbf{u}^i$ values with an unknown, latent value is nonsensical unless you _a priori_ know them. **And**, If you know them a priori, analysis of the post-ICA part **reduces** to interventional CRL: It suffices to investigate  the implicitly-known $\\{\mathbf{u}^i : i \in [n]\\}$ to ensure $\Phi$ to be a diagonal matrix.
> >
> > In the original submission, same connection was still made, only implicitly.
> >
> > We can always argue whether the permutation ambiguity is a big problem or not, that is not the point here. You are assuming that there are known interventional environments (i.e. data ftom $\mathbf{u}$ equal to some known $\mathbf{u}^i$). This is unfortunately NOT a generalization of the existing settings.
> >
> > I would like the authors to evaluate my argument and refute it if possible. _This_ is the reason I think the proposed novelty of the paper is incorrect.

---

> > > ### Author Response · Authors · 2025-11-24
> > >
> > > Thanks for your detailed comments and for helping us clarify the core point. We appreciate your careful reading and further interpretation. We now believe we are on the same page regarding whether the alignment of  $\mathbf{u}$ constitutes prior knowledge.
> > >
> > >
> > >
> > > First, we would like to highlight that we do **not** assume a priori alignment of the $\mathbf{u}$ values. Instead, this alignment is essentially a consequence of the identifiability of the latent noise variables up to linear scaling, which are recovered via nonlinear ICA. This identifiability result ensures that the $\mathbf{u}$ values are aligned across environments. We have added a paragraph (highlighted in teal) localized between Eq. (44) and Eq. (45), which explains why the function class of the estimated model must match that of the true model without assuming prior knowledge of $\mathbf{u}$ values.
> > >
> > > Specifically, for a given environment $\mathbf{u}_i$, Assumption (iv) implies that $z_i$ has no parent contribution in that environment, and thus $z_i = n_i$ under additive noise models. Suppose, toward a contradiction, that the estimated function $\hat h_i^{\mathbf{u}=\mathbf{u}_i}$ does not belong to the same function class, i.e., $\hat z_i$ has parent nodes. Then the corresponding $\hat z_i$ would necessarily mix at least two latent noise sources: its own $\hat n_i$ and the latent noise from parent nodes. This would violate the identifiability of $n_i$ (shown in Step I), leading to a contradiction ($z_i = n_i$ and thus should be identifiable, while $\hat z_i$ is a mxiture of two latent noises).
> > >
> > > In summary, the alignment of environment labels is **not assumed a priori**; rather, it is ensured by the identifiability result of latent noise variables via nonlinear ICA.  Without such alignment, the latent noise variables could not be identified. This is one of the key benefits of using nonlinear ICA for CRL.
> > >
> > > -----
> > >
> > > We hope this addresses your main concern.
> > >
> > > We sincerely appreciate your detailed discussion and comments.

---

> > > > ### Comment · Reviewer_nLhX · 2025-11-24
> > > >
> > > > Under the intervention $\mathbf{u}\_i$, we indeed have $z\_i = n\_i = \hat{n}\_i$ up to an affine transformation. It is also correct that $\hat{z}\_i = \hat{n}\_i + f(\hat{n}\_{\hat{p}a(i)})$ unless $\mathbf{u}\_i$ also is a hard intervention for $\hat{z}\_i$.
> > > >
> > > > My question is: why does the fact that $\hat{z}\_i$ is not solely a function of $\hat{n}\_i$ constitute a contradiction to $\hat{n}\_i = n\_i$ ? None of the preceding assumptions appear to require $\hat{z}\_i$ and $\hat{n}\_i$ to coincide structurally. Thus, even if $\hat{z}\_i$ depends on other noise variables, $n\_i = \hat{n}\_i$ could still hold. Importantly, even if for the "true" system $z\_i = n\_i$ holds, why are we supposing that it would also hold for any candidate system? This equivalence is precisely the statement that we are trying to prove that follows from _other_ assumptions and results, which makes this a circular argument.

---

> ### Author Response · Authors · 2025-11-24
>
> We sincerely thank the reviewer for the follow-up question and for the careful reading of our work.
>
> Importantly, the identifiability result from **Step I** demonstrates that the latent noise variable $n\_i$ is identifiable up to linear scaling via likelihood matching. Consequently, the estimated latent noise $\hat{n}_i$ obtained through likelihood matching is necessarily a linear transformation of the true $n_i$.
>
> Since both $z\_i$ and $\hat z\_i$ are linked through likelihood matching (Eq. (39)), this implies that if $z_i = n_i$ for the true system, then $\hat z\_i$ must be a linear scaling of $z_i$ (or $n_i$). Put differently, $\hat z_i$ cannot incorporate additional latent noise sources without contradicting the identifiability of $n_i$.
>
> We hope this clarifies the point, and we greatly appreciate your careful consideration and insightful comments.

---

> > ### Comment · Reviewer_nLhX · 2025-11-25
> >
> > Just to make sure, is Equation (39) $\mathbf{z} = \mathbf{\Phi}(\hat{\mathbf{z}})$? Because I do not see how "given $\mathbf{z} = \mathbf{\Phi}(\hat{\mathbf{z}})$ and $z\_i = n\_i$ for the true system" implies "$\hat{z}\_i$ must be a linear scaling of $z\_i$ (or $n\_i$)."
> >
> > I believe the author's confusion might be caused by the following: At $\mathbf{u} = \mathbf{u}\_i$ we have $z\_i = h\_i^{\mathbf{u}\_i}(\mathbf{n}) = n\_i$. We have $n\_i = \hat{n}\_i$ up to scaling from Step I. Does this imply we have identified $z\_i$ through $n\_i$? The answer is no in general, since we have not identified the mapping $h\_i^{\mathbf{u}}(\mathbf{n})$ for $\mathbf{u} \neq \mathbf{u}\_i$. In other words, there is nothing generalizable that is learnt between $n\_i$ and $z\_i$ that doesn't involve the special property of $\mathbf{u}\_i$ being a hard intervention.
> >
> > My point here is: There is nothing -- at least as far as I can see from this discussion -- that constrains the behavior of $\hat{\mathbf{z}}$ at $\mathbf{u}\_i$. The only constraint we can leverage explicitly depends on the property that the candidate system as well is a hard intervention at $\mathbf{u}\_i$ -- see above.
> >
> > In summary, $\hat{z}\_i$ is not identified just because $n\_i = \hat{n}\_i$ and $z\_i = n\_i$ at $\mathbf{u} = \mathbf{u}\_i$ -- that is, unless we know it should behave like the true system and have $\hat{z}\_i = \hat{n}\_i$ at $\mathbf{u} = \mathbf{u}\_i$ as well.

---

> ### Author Response · Authors · 2025-11-25
>
> Thank you for your further comments. We greatly appreciate your raising these concerns.
>
> **Q1:**. How "given $\mathbf{z} = \mathbf{\Phi}(\hat{\mathbf{z}})$ and $z_i = n_i$ for the true system" implies "$\hat{z}_i$ must be a linear scaling of $z_i$ (or $n_i$)."
>
> **R1:**
> Please note:
>
> According to Step I, we have $ n\_i = \hat{n}_i$ (E. 1).
>
> Given $\mathbf{z} = \mathbf{\Phi}(\hat{\mathbf{z}})$, we have $z\_i =  \mathbf{\Phi}\_i (\hat z\_{<i}, \hat z\_i)$ (E. 2).
>
> Accoding to Assumption (iv), we have $z_i = n_i$, for $\mathbf{u}_i$ (E. 3).
>
> Together with E. 2 and E. 3, we have $ n_i=  \mathbf{\Phi} (\hat z_{<i}, \hat z_i)$. (E. 4).
>
> Then, Combining  (E. 4) and (E. 1), we have $ \mathbf{\Phi}\_i(\hat z\_{<i}, \hat z\_i)=\hat n\_i$, which means that $\mathbf{\Phi}\_i(\hat z\_i)=\hat n\_i$ (here we have already shown $\hat z\_i$ must not depend on $\hat z\_{<i}$, at point $\mathbf{u}_i$. Then, applying this result to Eq. (44) gives $J\_{\mathbf{\Phi}\_{3,2}} = 0$.). For how $\mathbf{\Phi}\_i(\hat z\_i)=\hat n\_i$ reduce to $\hat z\_i=\hat n\_i$, please follow Eqs. (42) - (45).
>
>
> **Q2: there is nothing generalizable**
>
> **R2:** Please note, prior to Eq. (44), $\mathbf{u}$ is not specified. Thus, Eq. (44) holds for all $\mathbf{u}$. We then use a  specific choice $\mathbf{u}\_i$, as introduced in assumption (iv), to obatin $J\_{\mathbf{\Phi}\_{3,2}}=0$ as above. $J\_{\mathbf{\Phi}\_{3,2}}=0$ is generalizable.
>
> Thank you again for your comment, we look forward to your further feedback!
>
> Best

---

### Official Review · Reviewer_qMYX · 2025-11-04

**Soundness:** 3
**Presentation:** 3
**Contribution:** 2
**Rating:** 4
**Confidence:** 4

**Summary:**

The papers deal with the identification of latent causal models from high-dimensional observations and provide results for the case of general additive noise latent causal models, which extends over the prior results that were limited to linear or polynomial latent causal relationships (Liu et al. 2022, Liu et al. 2024). They utilize the commonly used assumption of an exponential family distribution conditioned on surrogate variables ($u$) for the noise variables ($n$), which helps them identify the noise variables up to permutation and scaling. Then they show how the identification guarantees on the noise variables can be used to get identification guarantees on the latent variables ($z$) themselves using the assumption that the causal influence of parent node does not contain any term invariant to surrogate variable ($u$). They benchmark the proposed approach on synthetic and semi-synthetic (fMRI) data from prior works against common baselines, and introduce a new real world human motion task.

**Strengths:**

- The identifiability results are interesting, building upon the prior works (Liu et al. 2022, 2024), they extend the identifiability of latent causal models with restrictive linear or polynomial assumptions, to more general additive noise causal mechanisms. The extensions for partial identifiability and post non-linear causal models is also significant and novel to the best of my knowledge.

- The paper is well-written with a clear description of the various assumptions needed for the theoretical results. I really like the intuitive explanations and examples offered behind the assumption 4 on the causal influences.

- The proposed approach is principled and the claims made in the paper are supported by experimentation over standard benchmarks from prior works. Also, the application to real-world human motion dataset is interesting. However, the scale of the experiments is limited (refer to the Weaknesses section below for more details).

**Weaknesses:**

- One main limitation of the work is that the authors don't mention in Section 3.1 that setup and assumptions 1-3 are very similar to the prior work by Liu et al. 2024 [1]. While the authors state that their novel contribution is only assumption 4 on causal influences, but the way the rest of the setup and assumptions are presented make it seem as if the its a contribution of this work, while the prior work Liu et al. 2024 works with the exact same formulation. It is weird that the authors compare against the prior work on latent polynomial causal model  identification in the experiments section, they do not explicitly mention this in their theoretical section. I suggest the authors to make major edits to Section 3.1 and make this connection explicit. Further this should be highlighted in the introduction as well.

- The scale (dimension of the latent variables) of the synthetic and semi-synthetic experiments is too small, with the maximum dimension of the latent space being 5. It would be nice to see the results with the proposed approach for larger latent dimensions ($d=10$ or $d=20$ at least).

- Given the emphasis on identifying general (latent) additive noise causal models, it would be nice to benchmark the proposed approach against the methods that identify linear (latent) additive noise models [2, 3].

*References*

[1] Liu, Yuhang, Zhen Zhang, Dong Gong, Mingming Gong, Biwei Huang, Anton van den Hengel, Kun Zhang, and Javen Qinfeng Shi. "Identifiable latent polynomial causal models through the lens of change." arXiv preprint arXiv:2310.15580 (2023).

[2] Squires, Chandler, Anna Seigal, Salil S. Bhate, and Caroline Uhler. "Linear causal disentanglement via interventions." In International conference on machine learning, pp. 32540-32560. PMLR, 2023.

[3] Jin, Jikai, and Vasilis Syrgkanis. "Learning linear causal representations from general environments: identifiability and intrinsic ambiguity." Advances in Neural Information Processing Systems 37 (2024): 63466-63509.

**Questions:**

Please refer to my the weaknesses section above for my major questions.

---

> ### Author Response · Authors · 2025-11-22
>
> We sincerely thank you for taking the time to review this draft.
>
> ----
>
> **Q1: Comparison with  Liu et al. 2024**
>
> **R1:** We thank the reviewer for raising this concern. We would like to clarify that:
>
> *Acknowledgement of Liu et al. 2024 in the main paper.*
>
> Liu et al. 2024 is explicitly discussed in the *Introduction*, *Related Work*, and throughout *all experimental comparisons* in the initial version. In these sections, we clearly state that Liu et al. 2024 studies *polynomial* latent causal models, whereas our work focuses on *general additive noise models*. We apologize if the connection was not sufficiently emphasized in Section~3.1.
>
> *Similarity of Assumptions 1-3 is natural and intended.*
>
> These assumptions are standard identifiability conditions in the nonlinear ICA literature as we claimed in the draft, and are shared by many existing works that leverage nonlinear ICA for causal representation learning, e.g., Liu et al. 2024, [1] and [2]. Since both approaches build upon the same nonlinear ICA foundation, it is expected that these assumptions take similar forms. We do not claim novelty for these assumptions.
>
>
> *Our theoretical novelty extends beyond the polynomial setting.*
>
> The key differences lie in:
>
> (i) Our model class covers general additive noise models, while Liu et al. 2024 is restricted to polynomial structural equations (which can be viewed as a special case of our formulation);
>
> (ii) Our Assumption (iv) introduces a non-parametric conditional mechanism to model changes in causal influences, whereas Liu et al. 2024 requires explicit changes in polynomial coefficients. This generalization necessitates substantially more general proof techniques than those used in the polynomial case.
>
> (iii) We also explore a potential application for CRL.
>
> *Revisions made following the reviewer’s suggestion.*
>
> We have made the following revisions:
>
> (i) added a new appendix subsection titled Section P, Relationship to Liu et al.\ (2024) providing a detailed comparison between the two works;
>
> ii inserted explicit cross-references in both Introduction and Sec. 3.1 pointing readers to this comparison;
>
> (iii) revised the text in Sec. 3.1 to state explicitly that Assumptions (i)-(iii) align with prior nonlinear ICA--based approaches, including Liu et al. 2024.
>
> We hope these clarifications and revisions address the reviewer’s concern and make the relationship between the two works fully transparent.
>
>
> [1] Kun Zhang, Shaoan Xie, Ignavier Ng, and Yujia Zheng. Causal representation learning from multiple
> distributions: A general setting. arXiv preprint arXiv:2402.05052, 2024.
>
> [2] Ignavier Ng, Shaoan Xie, Xinshuai Dong, Peter Spirtes, and Kun Zhang. Causal representation
> learning from general environments under nonparametric mixing. In International Conference on
> Artificial Intelligence and Statistics, 2025.
>
> **Q2: The scale (dimension of the latent variables) of the synthetic and semi-synthetic experiments is too small.. It would be nice to see the results with the proposed approach for larger latent dimensions (10 or 20 at least).**
>
> **R2:** Thanks for the suggestion. In fact, apart from the experiments on Real Human Motion with a latent dimension of 14, we had already evaluated our method on larger latent dimensions in the initial version. As reported in Section K of the Appendix, Figure~9 presents results for latent dimensions ranging from 6 to 15.
>
> **Q3:  Comparision with the methods that identify linear (latent) additive noise models [2, 3].**
>
> **R3:** We thank the reviewer for the suggestion. Linear latent additive-noise methods [2,3] are not included in our comparisons because their structural assumptions are fundamentally incompatible with the models and assumptions studied in our paper.
>
> Specifically, these methods assume a linear mapping from latent to observed variables. In contrast, our theoretical results concern nonlinear latent-to-observed transformations. As a result, applying linear methods to our nonlinear data-generating processes would be guaranteed to fail by construction, not due to method weakness, but because their underlying assumptions are violated. A linear unmixing process cannot invert a nonlinear mapping. Moreover, there are additional incompatibilities: for example, [2] requires single-node interventions, and [3] requires that at most one component is Gaussian. As a result, such comparisons would be scientifically misleading and not informative, as it involves a fundamental model mismatch.
>
> Instead, we benchmark against baselines that share as many structural assumptions as possible, including the closest prior work (e.g., \citet{liu2023identifiable}), thus providing as meaningful an empirical comparison as possible.
>
> We hope this clarifies why benchmarking against [2,3] is neither feasible nor informative under our setting.

---

### Official Review · Reviewer_2wYV · 2025-11-05

**Soundness:** 2
**Presentation:** 3
**Contribution:** 4
**Rating:** 2
**Confidence:** 4

**Summary:**

This paper presents theoretical conditions for identifiability of CRL with latent additive noise models. The assumptions are standard in recent advances in nonlinear ICA and generative models. The authors accompany the results with estimation based on variational inference, and present synthetic ablations along with interesting applications of their setup to real-world settings.

**Strengths:**

- The document presents thorough intuitions for the modelling choices and assumptions. This makes the whole manuscript clear in terms of readability.
- Investigating partial identifiability for a subset of failed assumptions is a very interesting contribution.
- The experiments are thorough and the real-world showcase is exciting.

**Weaknesses:**

**Comments on theoretical analysis:**
- **Clarity:** Please add brief proof sketches after the main theorems: A few indications each explaining the proof idea, how tools from nonlinear ICA/aux-variable methods are used, and explicit pointers to the exact appendix sections/lemmas.
- **Sufficiency and necessity:** The paper mentions that their conditions are both **sufficient and necessary**. However, necessity here is considered under their specific model restrictions and assumptions, which is a very specific setup that requires auxiliary variables. I believe this statement should be removed as it might suggest that without these assumptions, CRL is not possible for latent ANMs.
  - Line 71: “Notably, our analysis shows that the proposed nonparametric condition is both necessary and sufficient for identifiability under the nonlinear ICA framework, without additional constraints.”
  - Line 108: “a sufficient and necessary condition for changes in the causal influences”
  - For alternative solutions, [1] uses a polynomial model, which is a sufficient condition for CRL with ANMs.
- Section 3.3 cites PNL for a two-variable case. Would it be possible to cite a multivariate version of this work (if it exists)?

**Theoretical concerns**

I checked the proofs and I have the following concerns:
  - Proof of Theorem 3.1
    - The proof introduces h, but it has not been defined anywhere in the main text or in the proof. I can see it defined in Line 897, but it’s quite hidden in the Appendix. Consider introducing this notion in the main text or your theorem proof before writing $\hat{\mathbf{h}}$ and $\mathbf{h}$. A small paragraph with this notion will significantly improve clarity.
    - **Critical issues:** The following requires major revision. Step I of this proof requires some invertible $\hat{f}$, which recovers some latents $\hat{z}$. For this, the authors use Lemma B.1, where they **assume there exists $\hat{f}$ which recovers some latents**. There are several issues with this point.
      - This Lemma is only referenced once at the end of the proof of Theorem 3.1.
      - The proof itself introduces a **new assumption**, which is very strong and I am not sure this holds in general.
      - The reason why it does not hold is because we have $\dim(\hat{z}) = \dim(z) + \dim(\varepsilon)$. However, in the proof, the authors compose $\hat{f}$ with some $\hat{h}$, which is prohibited due to dimensionality mismatch.
      - I believe the authors aim to replicate the strategy from GIN [2]. However, in their paper, they establish equivalence with respect to some latent variables with potentially higher dimensions. Please, see Equation (9) from Sorrenson et al. (2020).
      - I can recommend a very quick solution to this, which is to revert back to iVAE’s setup [3] and do the convolution trick to remove the noise. It’s less flexible but quite reasonable as you base your estimation on VAEs and not normalising flows.
    - Line 1054: “For simplicity, in the following we neglect the noise term $\varepsilon$”.
      - Given the above point, you can only neglect the noise term if you are able to disentangle it from your system. However, $\hat{z}$’s dimensionality contains $d - \ell$ dimensions which are noise. Resolving the above problem should also fix this issue (you can then just remove this sentence).
    - Line 1119: “we can naturally use this as prior knowledge to impose the same constraint on the inference model”.
      - This might be confusing to some readers. The correct phrasing should be that both models belong to the same model class, so you can just say that the partial derivatives in both cases come from assumption (iv). Mentioning inference and generative model is confusing, as we want to think of identifiability as the same distribution but for two generative models (with distinct $h$, $\hat{h}$ under the same model class).
 - Lemma B.2 claims: "determinant = 1" $\to$ "h is invertible". However, I believe this is imprecise as the invertibility comes from construction of the additive noise model structure where you have $n_i = z_i - g_i(pa_i)$. Please revise the last sentence in Line 917.

The rest of the theory looks alright to me with enough details. I would be glad to re-assess my score if you resolve my comments above.



**Estimation comments**
- Please note explicitly that the objective you use is the ELBO, indicate it is a lowerbound of the log-likelihood, and make a small note on estimator consistency. A reference to [3] and the intuitions for assumptions with some relfections on your specific setup would be ideal.

**Missing Baselines**
- Would it be possible to include CausalVAE [4] as a baseline?

[1] Liu, Yuhang, et al. "Identifiable latent polynomial causal models through the lens of change." arXiv preprint arXiv:2310.15580 (2023).

[2] Sorrenson, Peter, Carsten Rother, and Ullrich Köthe. "Disentanglement by nonlinear ica with general incompressible-flow networks (gin)." arXiv preprint arXiv:2001.04872 (2020).


[3] Khemakhem, Ilyes, et al. "Variational autoencoders and nonlinear ica: A unifying framework." International conference on artificial intelligence and statistics. PMLR, 2020.

[4] Yang, Mengyue, et al. "Causalvae: Structured causal disentanglement in variational autoencoder." arXiv preprint arXiv:2004.08697 (2020).

**Questions:**

- Would it be possible to use a mixture model to relax the requirement of auxiliary variables?
- I have some experience with VAEs and structure learning and generally the MCCs have a sharp decrease when increasing latent dimensions. How does your approach scale to higher dimensions? (I am not asking for additional experiments)
- Is MCC (mean coefficiant correlation) from standard ICA literature the same as MPC?

---

> ### Author Response · Authors · 2025-11-22
>
> Before response, we would like to sincerely thank you for the exceptionally careful and detailed evaluation of our work. We truly appreciate the substantial time and effort spent checking our theoretical proofs end-to-end. We respect the reviewer’s rigor, and their comments have helped us significantly improve the clarity and precision of the paper.
>
> ----
>
> **Q1: Please add brief proof sketches after the main theorem**
>
> **R1:** We fully agree with the reviewer’s suggestion and have added concise proof sketches after each main Theorem and Corollary.
>
> **Q2: the statement "sufficiency and necessity" should be removed**
>
> **R2:** We agree with your concern and have removed the statements in the revised version, since the necessity only holds under the specific model assumptions and could be misleading if interpreted more broadly. Related parts have also been revised to clarify that the conditions should be interpreted as sufficient.
>
> **Q3: PNL for multivariate case**
>
> **R3:** There do exist works extending PNL to the multivariate setting (e.g., [1,2]). We have added these references in the revised version to provide a more complete view and strengthen the discussion of PNL models. Thanks a lot.
>
> **Q4: The defination of h, and  a small paragraph with notation before proof**
>
> **R4** We have provided a definition of $\mathbf{h}$ at the beginning of the lemma (see Definition 1 for details). Additionally, both
> $\mathbf{h}$ and the related $\mathbf{\hat h}$ are explicitly introduced and notated at the beginning of the proof to improve clarity. We agree that this enhances clarity and thanks for the suggestion.
>
> **Q5: For the Critical issues regarding $\mathbf{\hat f}$**
>
> **R5:** We sincerely thank the reviewer for pointing this out and for kindly suggesting the solution. Following this advice, we have carefully revised the related parts to address the concern, clarifying the assumptions and the construction in Step I. Specifically, these modifications include modeling the noise in Eq. 3 explicitly as additive, updating the corresponding Assumption (i), and, most importantly, modifying Step I itself. We believe these changes substantially improve the rigor and clarity of the argument. Also, With these revisions, the previously problematic sentence, `For simplicity, in the following we neglect the noise term', has been gently sent into early retirement...... Thanks a lot!
>
> **Q6: Line 1119: “we can naturally use this as prior knowledge to impose the same constraint on the inference model”**
>
> **R6:** We agree that mentioning the inference and generative models together may be confusing. After much internal debate (it turns out simplicity really is the best way to avoid confusion......), we have revised this part for clarity, yes, as you suggested, it is just `both models belong to the same model class'. Thanks again!
>
> **Q7: Lemma B.2 claims: "determinant = 1" ->"h is invertible". However, I believe this is imprecise**
>
> **R7:** We thank the reviewer for pointing this out. We have revised the last sentence in Lemma B.2 to clarify that the invertibility of h follows from the construction of the additive noise model (ANM) structure (Yes,.. the cause is the construction of ANM :)...)
>
> **Q8: Estimation comments, EBLO, a small note on estimator consistency**
>
> **R8:** Thank you for the helpful comment. We have updated the manuscript to explicitly clarify that our objective is the evidence lower bound (ELBO) of the log-likelihood, included a note on estimator consistency, and provided a discussion linking the proposed VAE method to our model assumptions.
>
>
> **Q9: Would it be possible to include CausalVAE [4] as a baseline?**
>
> **R9:** We include CausalVAE [4] as a baseline for the Semi-Synthetic fMRI data. The results can be found in Figure 3 (a).  In this experiment, CausalVAE consistently produces fully connected graphs across all random seeds.
>
> [1] Uemura, Kento, et al. "A multivariate causal discovery based on post-nonlinear model." Conference on Causal Learning and Reasoning. PMLR, 2022.
>
> [2] Keropyan, Grigor, David Strieder, and Mathias Drton. "Rank-based causal discovery for post-nonlinear models." International Conference on Artificial Intelligence and Statistics. PMLR, 2023.

---

> ### Author Response · Authors · 2025-11-22
>
> **Q10: Would it be possible to use a mixture model to relax the requirement of auxiliary variables?**
>
> **R10:** We truely appreciate the opportunity to share our thoughts on this question.
>
> We agree that mixture-based formulations offer a promising path toward relaxing the need for auxiliary variables, and we view this as an important direction for future work. On the one hand, prior work suggests that, in general, inferring auxiliary variables is impossible without imposing additional assumptions [3] (Inferring auxiliary variables in an unsupervised manner, in general, seems to be impossible). On the other hand, recent progress in [4] demonstrates that nonlinear ICA can be made identifiable by leveraging the identifiability of Gaussian mixture models (GMMs) in the observed space.
>
> Taken together, this “surface-level conflict” raises several open questions. For instance, to what extent can the identifiability properties of GMMs help circumvent the impossibility results highlighted in [3]? Furthermore, when shifting the focus from nonlinear ICA to latent causal variables, it remains unclear how mixture-based probabilistic models, potentially extending beyond GMMs, such as mixtures of exponential families [5], interact with the structural causal model framework. These questions are both interesting and challenging, and we consider them promising directions for future exploration.
>
> ***Q11: I have some experience with VAEs and structure learning and generally the MCCs have a sharp decrease when increasing latent dimensions. How does your approach scale to higher dimensions?**
>
> **R11: You raise another excellent point!**
>
> Our approach indeed faces similar challenges when scaling to higher latent dimensions. As shown in Figure 9, the MPC decreases as the latent dimension grows, and the SHD becomes unfavorable (as shown in the right in Figure 1, the SHD is already unsatisfactory even for relatively small latent dimensions). Our current thought is that this degradation might stems from the architecture of the inference model: we first attempt to recover the noise variables $\mathbf{n}$ via an encoder (We note that some prior works have also done so), and then learn the causal structure over latent causal variables $\mathbf{z}$. However, once the encoder (usually implemented by a MLP) is tasked with recovering $\mathbf{n}$, it may implicitly need to learn aspects of the causal structure among $\mathbf{z}$ (viewed as $x \to z \to n$ in inference). The question is, in practice, it may be challenging for an MLP-based encoder to reliably capture a discrete DAG structure.
>
>
> **Q12: Is MCC (mean coefficiant correlation) from standard ICA literature the same as MPC?**
>
> **R12:** We understand that many nonlinear ICA papers report using MCC as an evaluation metric. In implementation, MCC is often computed in two ways:
>
> Pearson correlation: Measures linear correlation and is consistent with MPC.
>
> Nonlinear correlation: Used in identifiability results up to a nonlinear transformation (e.g., Gaussian latent variables with environment-dependent variance in nonlinear ICA).
>
> Since our theoretical results guarantee identifiability up to linear scaling, MPC is more appropriate, as it directly aligns with our theoretical guarantees. In contrast, using a nonlinear correlation measure is not suitable in this setting, as it may produce inflated or misleading MCC values that do not accurately reflect identifiability under linear scaling.
>
>
> [3] Lin, Yong, et al. Zin: When and how to learn invariance without environment partition? NeurIPS 2022.
>
> [4] Kivva, Bohdan, et al. Identifiability of deep generative models without auxiliary information. NeurIPS 2022.
>
> [5] Teicher, Henry. Identifiability of Mixtures of Exponential Families. J. Math. Anal. Appl., 1965.

---

### Meta-Review · Area_Chair_B5qx · 2026-01-02

**Summary:**

This paper considers causal representation learning with latent additive noise models. On theory side, it provides identifiability results based on assumptions related to nonlinear ICA and identifiable generative models. On the algorithmic side, it provides a learning algorithm based on VAEs. Experiments on synthetic and real-world tasks (human motion dataset) show the efficacy of the approach.

The AC understands that this paper targets for both theoretical and practical contributions. Although questions regarding experiments and baselines are largely addressed, questions still remain for novelty as well as the theoretical proofs (see below boxes).

In the rebuttal and revision process, the theory part of the paper has been revised substantially. However, and unfortunately, due to the OpenReview incident, the revised theory presentation did not receive sufficient scrutiny for correctness and novelty re-assessment (see below boxes). The AC thinks it is inappropriate to accept a theory-heavy paper without rigorous reviews on the main theoretical result.

**Reviewer Concerns:**

Questions addressed:
- Experimental comparison with further baselines (e.g., CausalVAE).

Questions outstanding:
- Proof of Theorem 3.1 (a major result of the submission). Revision has modified one of the assumptions and the proof so this is a substantial edit. Unfortunately due to OpenReview incident this revision has not received further review in a proper way.
- Novelty of the theoretical result, still centred around Theorem 3.1. The key debating point is whether the method requires the $u_i$ environment variables to be known. Reviewer nLhX argued that the proposed theory needs known $u_i$ and therefore the theoretical framework is still closely related to previous non-linear ICA type results (e.g., iVAE) hence less novelty.

The AC's assessment regarding the novelty issue: in revision the authors provided more details for Theorem 3.1 proof which includes very similar strategies as in the identifiability proof of iVAE. Since in algorithmic implementations iVAE uses known $u_i$ variables, effectively this is assuming known $u_i$ and therefore the AC supports Reviewer nLhX's argument.

**Reviewer Scores:**

The major concerns from reviewers are about theory, mainly from reviewers Reviewer 2wYV and Reviewer nLhX. I don't think they would have increased their score after author rebuttal as they would have thought the theory wasn't presented clearly and was relying on standard assumptions in previous work (therefore less novelty).

---

### Decision · Program_Chairs · 2026-01-26

Reject